# Spectral Estimation with Free Decompression

**Siavash Ameli**
ICSI and Department of Statistics
University of California, Berkeley
sameli@berkeley.edu

**Chris van der Heide**
Dept. of Electrical and Electronic Eng.
University of Melbourne
chris.vdh@gmail.com

**Liam Hodgkinson**
School of Mathematics and Statistics
University of Melbourne
lhodgkinson@unimelb.edu.au

**Michael W. Mahoney**
ICSI, LBNL, and Department of Statistics
University of California, Berkeley
mmahoney@stat.berkeley.edu

## Abstract

Computing eigenvalues of very large matrices is a critical task in many machine learning applications, including the evaluation of log-determinants, the trace of matrix functions, and other important metrics. As datasets continue to grow in scale, the corresponding covariance and kernel matrices become increasingly large, often reaching magnitudes that make their direct formation impractical or impossible. Existing techniques typically rely on matrix-vector products, which can provide efficient approximations, if the matrix spectrum behaves well. However, in settings like distributed learning, or when the matrix is defined only indirectly, access to the full data set can be restricted to only very small sub-matrices of the original matrix. In these cases, the matrix of nominal interest is not even available as an implicit operator, meaning that even matrix-vector products may not be available. In such settings, the matrix is "impalpable," in the sense that we have access to only masked snapshots of it. We draw on principles from free probability theory to introduce a novel method of "free decompression" to estimate the spectrum of such matrices. Our method can be used to extrapolate from the empirical spectral densities of small submatrices to infer the eigenspectrum of extremely large (impalpable) matrices (that we cannot form or even evaluate with full matrix-vector products). We demonstrate the effectiveness of this approach through a series of examples, comparing its performance against known limiting distributions from random matrix theory in synthetic settings, as well as applying it to submatrices of real-world datasets, matching them with their full empirical eigenspectra.

## 1 Introduction

Eigenvalues encode essential information about matrices and are central to the evaluation of spectral invariants in computational linear algebra. Two such invariants, the log-determinant and the trace of the inverse, appear frequently in statistical modeling and Bayesian inference (Ubaru & Saad, 2017). However, these quantities require access to the full range of eigenvalues, posing substantial challenges in modern large-scale settings. As datasets continue to scale up, it is becoming increasingly common to encounter matrices that cannot be practically formed in their entirety. This may occur, for example, if the matrices are so large that they exceed the storage capacity in local memory. Popular methods for operating with large matrices often rely on randomized low-rank approximations from Randomized Numerical Linear Algebra (RandNLA) (Drineas & Mahoney, 2016; Murray et al., 2023; Dereziński & Mahoney, 2021), and can broadly be classified as either iterative subspace or sampling methods, although hybrid methods have become prominent (Ubaru et al., 2017; Saibaba et al., 2017; Chen

et al., 2021). *Implicit methods* (Adams et al., 2018) that rely on Krylov subspaces using Arnoldi- or Lanczos-based techniques (Ubaru et al., 2017) have been successful for estimating arbitrary spectral invariants (Gardner et al., 2018; Potapczynski et al., 2023), and they require access only to matrix-vector products. These techniques avoid explicit matrix storage and can often converge quickly. However, their performance degrades significantly when the matrix is ill-conditioned.

The situation is *particularly dire* when most matrix entries are inaccessible and matrix-vector products are unavailable or too costly. This can occur when the data used to generate the matrix is spread across a distributed system, or when the matrix itself is enormous (see e.g., Ameli et al. (2025)). Such matrices can be considered *impalpable*, because neither the matrix nor its matrix-vector products can be handled in their entirety. Despite being inaccessible, the information these impalpable matrices contain can be crucial, making methods to summarize or access them highly valuable. Approaches which subsample a few rows and columns are feasible, assuming that any observed quantities can be directly extended to the full matrix (Drineas et al., 2008). This is the strategy behind the Nyström method, which accesses a matrix via column/row sampling, and then reconstructs the matrix under linearity assumptions (Gittens & Mahoney, 2016; Gittens et al., 2016). Unfortunately, such techniques incur significant bias, as they target behavior dictated by large eigenvalues by design, giving less prominence to the near-singular dimensions that play a crucial role in determinants and inverses.

In the setting of impalpable matrices, or when otherwise fine-grained behavior of ill-conditioned matrices becomes important and implicit methods fail, there are very few tools currently available (Couillet & Liao, 2022; Ameli et al., 2025). To address this gap, we provide a novel method for approximating the empirical spectral density of extremely large (impalpable) matrices, requiring only access to appropriate submatrices. Our technique is based on tools from random matrix theory and free probability, and it applies to virtually any class of matrices. Assuming only that the underlying matrix is sufficiently large, the empirical spectral density of a (sampled) submatrix can be evolved by a partial differential equation (PDE) in terms of the dimension of the matrix to access the spectral density of the full (impalpable) matrix. Due to connections with free probability theory, which we explain below, we call this process **free decompression**.

To summarize, we:

(I) introduce and derive *free decompression*, the first method for spectral density estimation of impalpable matrices, by evolving the empirical spectral density of submatrices, and provide rigorous polynomial error bounds under mild assumptions (Proposition 4);

(II) demonstrate the utility of our free decomposition method for a range of synthetic examples, social media graph data, and empirical neural tangent kernels; and

(III) provide a user-friendly Python package, `freealg`,[1] implementing all proposed algorithms, that reproduces the numerical results of this paper.

The remainder of this paper is structured as follows: Section 2 provides background and outlines the different classes of matrices of interest; Section 3 provides the necessary tools in order to construct our free decompression procedure; Section 4 describes its implementation and issues faced in operationalization; Section 5 provides numerical examples. We conclude with discussion in Section 6. A summary of the notation used throughout the paper is provided in Appendix A. An overview of free decompression and its application to well-studied random matrix ensembles is presented in Appendix B.1. Discussions of the significant numerical considerations required to perform free decompression are provided in Appendix C and Appendix D. Proofs of our theoretical contributions can be found in Appendices E and F. Background on the neural tangent kernel example is provided in Appendix G. Finally, Appendix H presents a guide to our implementation of free decompression.

## 2 Impalpable Matrices

We are interested in estimating the spectral density of a wide class of matrices for which operating on the full matrix is infeasible. In the most extreme case, these matrices cannot be formed in their entirety, either implicitly or explicitly. We call these matrices *impalpable*. We will consider the case where the impalpable matrix of interest is Hermitian. Since we are interested in the setting where we

---

[1]The source code of the Python package `freealg` is available at https://github.com/ameli/freealg, with documentation at https://ameli.github.io/freealg.

cannot access the full matrix, we will work with submatrices indexed by randomly sampled columns of the impalpable matrix of interest, where the submatrix consists of the intersection of the impalpable matrix's rows and columns that appear in this index set. This corresponds to an assumption that the matrices are part of a sequence that is *asymptotically free* (Maïda, 2023). We will then see that it is convenient to work with tools from random matrix theory and free probability, namely the familiar Stieltjes transform (Couillet & Liao, 2022), whose favorable properties we exploit in our free decompression procedure.

## 2.1 Categories of Matrix Difficulty

We find it useful to classify matrices by four modes of computational difficulty: explicit, implicit, out-of-core, and impalpable, the latter of which is our focus.

**Explicit Matrices.** This is the most straightforward class of matrices. They can be formed explicitly as contiguous arrays and operated on in memory. Existing implementations on such matrices are highly optimized, with their runtime limited only by their computational complexity.

**Implicit Matrices.** Difficulties begin to arise once matrices become too large to be stored in memory. *Implicit matrices* are those that can be computed or estimated only in terms of matrix–vector products $v \mapsto \mathbf{A}v$ (Adams et al., 2018). Treating a matrix $\mathbf{A} \in \mathbb{R}^{n \times n}$ as implicit often has significant computational benefits, even before the memory wall (Gholami et al., 2024); for iterative methods, computation time can often be reduced, e.g., from $\mathcal{O}(n^3)$ operations to $\mathcal{O}(n^2)$. If the formation of the matrix–vector product is efficient, this can be further reduced to $\mathcal{O}(n)$. Memory requirements may follow similar reductions. Implicit methods exist for most standard linear algebra operations, often based on Arnoldi- or Lanczos-based iterations, and appear in mature software packages (Potapczynski et al., 2023). One of the most popular implicit matrix algorithms for spectral function estimation is *stochastic Lanczos quadrature* (SLQ) (Ubaru et al., 2017); however, the performance of such methods may deteriorate for highly ill-conditioned matrices (Ameli et al., 2025).

**Out-of-Core Matrices.** Implicit methods typically operate on Krylov iteration schemes, with error rates depending on condition numbers of the matrix—see Bhattacharjee et al. (2025), e.g., in the case of SLQ. This can be disastrous for large and ill-conditioned dense matrices, with accurate estimates requiring a large number of matrix–vector products. Exact methods become necessary in this setting if a high level of accuracy is required, but the memory bottleneck still remains. *Out-of-core* matrices have the property that for some integer $m \ll n$, any subblock of size $\mathbb{R}^{m \times m}$ can be formed explicitly and operated on in memory, and the full matrix can be stored locally by iterating through these blocks (Aggarwal & Vitter, 1988). Decompositions (Rabani & Toledo, 2001; Mu et al., 2014), inverses (Caron & Utard, 2002), and products of out-of-core matrices can be (slowly) computed to high precision, assuming adequate (typically very large) available external storage.

**Impalpable Matrices.** The most extreme category of matrices includes cases that do not fit into the other categories. Here, the matrix cannot be formed in memory, as it is either too large for out-of-core operations, or has missing portions. Matrix-vector products are assumed too expensive to compute accurately, or are rendered useless due to strong ill-conditioning. In principle, such a matrix can be considered "impalpable," as the vast proportion of either its elements or properties cannot be operated on. Comprising the worst-case

| | **Access in Memory** | | |
| **Type** | Matrix | Matrix–Vector Product | Any Subblock |
| --- | --- | --- | --- |
| Explicit | ✓ | ✓ | ✓ |
| Implicit | ✗ | ✓ | ✗ |
| Out-of-core | ✗ | ∼ | ✓ |
| Impalpable | ✗ | ✗ | ✗ |

Table 1: Comparison of matrix classes, according to attributes that can be formed and operated on in memory.

scenario, impalpable matrices offer unique challenges for linear algebra practitioners. Determining accuracy of any proposed solution is an underdetermined problem in general. The only feasible strategy is *extrapolation*, using available information. While this is inherently risky, the class of impalpable matrices simply offer no alternative. To our knowledge, only the FLODANCE algorithm of Ameli et al. (2025) offers a valid approach for estimating log-determinants of impalpable matrices.

## 2.2 Examples of Impalpable Matrices

**Enormous Gram Matrices.** Our motivating example of an impalpable matrix is an ill-conditioned Gram matrix $\mathbf{A} = (\kappa(\boldsymbol{x}_i, \boldsymbol{x}_j))_{i,j=1}^n$ comprised of a massive number $n$ of points $\boldsymbol{x}_i$ and a kernel $\kappa$ that is expensive to compute. In this case, computing a few subblocks of $\mathbf{A}$ is viable, but out-of-core methods are too expensive. Ameli et al. (2025) demonstrates the challenges of computing the log-determinant of a *neural tangent kernel matrix* for a large neural network model. Their most cost-effective strategy extrapolated the behavior of the log-determinant from smaller submatrices in accordance with a scaling law. This is an example of an *impalpable matrix strategy*, relying on smaller portions of the matrix without matrix–vector products. The potential gains are best demonstrated with a synthetic example: the covariance matrix of $\boldsymbol{x}_1, \ldots, \boldsymbol{x}_n$, where each $\boldsymbol{x}_i \sim \mathcal{N}(\mathbf{0}_p, \mathbf{I}_{p\times p})$ is $p$-dimensional. Once $p$ and $n$ become extremely large, forming this matrix is prohibitive. However, its spectral density is well-approximated by the Marchenko–Pastur law (Marčenko & Pastur, 1967), and its log-determinant over submatrices of increasing size is predictable (Nguyen & Vu, 2014; Cai et al., 2015; Ameli et al., 2025; Hodgkinson et al., 2023a).

**Catastrophic Cancellation.** Impalpable matrices typically arise due to issues of scale, but they need not be large themselves. Indeed, such matrices can arise due to *catastrophic cancellation*. Some matrices can be comprised of a non-trivial product of rectangular matrices; for example, $\text{cond}(\mathbf{A}^\mathsf{T}\mathbf{A}) = \text{cond}(\mathbf{A})^2$, so representing the outer product $\mathbf{A}^\mathsf{T}\mathbf{A}$ (e.g. in the normal equations) in floating-point precision can incur greater numerical error (Ameli et al., 2025). Higham (2022) refers to the formation of $\mathbf{A}^\mathsf{T}\mathbf{A}$ as a "cardinal sin" of numerical linear algebra. More products yield even larger errors. General Gram matrices exhibit similar behavior: the condition number can become so large that the matrix cannot be accurately represented in double precision, rendering even exact methods hopeless (Lowe et al., 2025). Fortunately, submatrices have smaller condition numbers, so there may be a submatrix that *can* be represented in double precision. Ideally, spectral behavior of these submatrices can then be extrapolated to obtain insight into the behavior of the full matrix in a meaningful way.

## 3 Free Decompression

Our objective is to approximate the spectral distribution of an impalpable matrix $\mathbf{A}$, given only explicit access to a randomly sampled $n \times n$ submatrix. This seemingly intractable task can be accomplished by working with the Stieltjes transform of the submatrix, and observing that for large $n$, this Stieltjes transform varies with $n$ in a manner that is well approximated by a PDE.

> **Free decompression** of a random submatrix $\mathbf{A}_n$ to a larger matrix $\mathbf{A}$ requires:
>
> 1. **estimation** of its Stieltjes transform $m_{\mathbf{A}_n}$;
>
> 2. **evolution** of $m_{\mathbf{A}_n}$ in $n$ via equation (1);
>
> 3. **evaluation** of the spectral distribution of $\mathbf{A}$.

This enables an approximation of the Stieltjes transform of the full matrix $\mathbf{A}$, from which a corresponding spectral distribution is easily computed. We emphasize that the matrix $\mathbf{A}$ itself may be arbitrary and deterministic. Asymptotic freeness enters our model through the method of sumbatrix selection, making it amenable to random matrix theoretic and free probabilistic tools.

### 3.1 Tools from Free Probability

In order to make free decompression concrete, we require tools from free probability. Let $\mathbf{A}$ be a fixed matrix of large size. If we can approximate the spectral density of a small randomly sampled submatrix, and extrapolate its behavior to the full matrix, it should also be possible to extrapolate *further* to larger matrices than the one we are given. To this end, we treat $\mathbf{A}$ as an element in an infinite sequence of matrices $\{\mathbf{A}_1, \mathbf{A}_2, \dots\}$ of increasing size, where $\mathbf{A}_n \in \mathbb{R}^{n\times n}$. For consistency, we assume that for each $n$, the matrix $\mathbf{A}_n$ is the top-left $n \times n$ submatrix of $\mathbf{A}_{n+1}$. This ensures that $[\mathbf{A}_n]_{ij} = [\mathbf{A}]_{ij}$ is constant in $n$ for fixed $i, j \leq n$. We will also assume that for any integer $k \geq 1$, the quantity $\frac{1}{n}\text{tr}(\mathbf{A}_n^k)$ converges as $n \to \infty$. This is equivalent to assuming that the empirical spectral distributions of $\mathbf{A}_n$ converge weakly as $n \to \infty$ (Voiculescu, 1991). Note that this imposes no requirements on the matrix to be approximated.

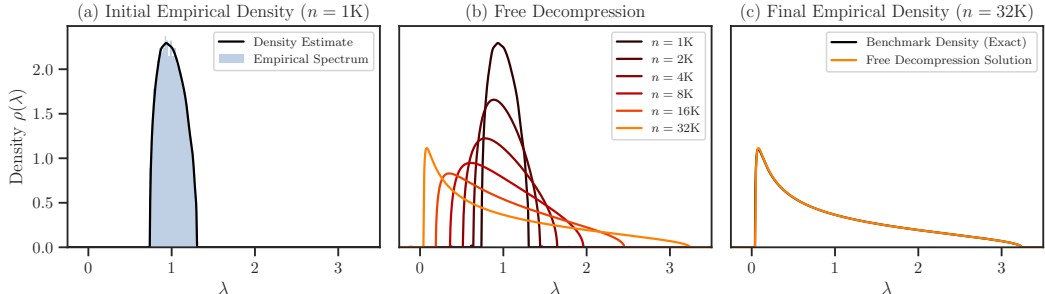

Figure 1: (a) Estimation of the Marchenko–Pastur law with ratio $\lambda = \frac{1}{50}$ from eigenvalues of Wishart matrices with $d = 50,000$ degrees of freedom and size $n \times n = 1000 \times 1000$; (b) approximation of densities for various values of $n$ via free decompression (Algorithm 1); and (c) comparison of our decompressed approximation of $n = 32,000$ with the known limiting analytic density for $\lambda = \frac{32}{50}$.

Still, we cannot predict the upcoming row and column in $\mathbf{A}_{n+1}$ from $\mathbf{A}_n$ alone, and we will need to appeal to random transformations to connect each element in the sequence together. One way to transform $\mathbf{A}$ into a random matrix, while preserving its eigenspectrum, is to perform a similarity transform with a random matrix $\mathbf{P}$, recovering $\mathbf{P}\mathbf{A}\mathbf{P}^{-1}$. The most inexpensive choice of random matrix $\mathbf{P}$ is a *(Haar) random permutation matrix*, so that the (random) similarity transform is equivalent to the following operation: let $\mathcal{S}_n$ be the uniform distribution on the space of permutations on $\{1, \ldots, n\}$, and consider the matrix $\mathbf{A}_n^\sigma = (\mathbf{A}_{\sigma(i)\sigma(j)})_{i,j=1}^n$, where $\sigma \sim \mathcal{S}_n$.

**Free Compression.** Due to a result of Nica (1993), the sequence of matrices $\mathbf{A}_n^\sigma$ is asymptotically free as $n \to \infty$, in the sense of free probability theory. We shall not further discuss free random matrices here, but we refer the interested reader to Maïda (2023) for a detailed introduction. Instead, we rely on one useful property.

Let $m_\mathbf{A}(z)$ denote the Stieltjes transform of $\mathbf{A}$ of size $n \times n$, given by $m_\mathbf{A}(z) = n^{-1}\mathbb{E}[\mathrm{tr}(\mathbf{A} - z\mathbf{I})^{-1}]$, and let $\omega(z)$ denote the functional inverse of $m(z)$, satisfying $m(\omega(z)) = z$. The corresponding $R$-transform is then defined as $R(z) = \omega(-z) - z^{-1}$. Our method depends on the following fundamental theorem of Nica & Speicher (1996); see also Olver & Nadakuditi (2012, Section 7).

**Theorem 1** (NICA-SPEICHER)**.** *For a free random matrix $\mathbf{A}$ of size $n \times n$ with $R$-transform $R(z)$, the $R$-transform of the top-left $n_s \times n_s$ submatrix of $\mathbf{A}$ is given by $R_{n_s}(z) = R(z\frac{n_s}{n})$.*

In other words, to arrive at the $R$-transform for a submatrix comprised of only the top-left proportion $\alpha$ of rows and columns, one need only scale the argument by $\alpha$. In free probability theory, this procedure is called *free compression* (Olver & Nadakuditi, 2012).

**Free Decompression.** Our main approach is based on the observation that nothing obstructs this operation from being conducted *in reverse*, to glean information about a larger matrix from a smaller one. For this reason, we call our procedure *free decompression*: starting from the $R$-transform of a smaller $n_s \times n_s$ submatrix $R_{n_s}(z)$, the $R$-transform of the larger $n \times n$ matrix is given by $R_n(z) = R_{n_s}(\frac{n}{n_s}z)$.

**Example** (COVARIANCE MATRICES)**.** Let $\mathbf{X} \in \mathbb{R}^{n \times d}$ have elements $X_{ij} \sim \mathcal{N}(0, 1)$ and let $\mathbf{A} := \frac{1}{d}\mathbf{X}\mathbf{X}^\mathsf{T} \in \mathbb{R}^{n \times n}$. Assuming $d \geq n$, then $\mathbf{A}$ is almost surely full rank, and it is $n \times n$ Wishart-distributed with $d$ degrees of freedom. The top-left $n_s \times n_s$ corner of $\mathbf{A}^\sigma$ is also Wishart-distributed with $d$ degrees of freedom; and so, for large $n_s$ and $d$, its eigenvalues approximately follow the Marchenko–Pastur law with ratio $\lambda = n_s/d$. The $R$-transform is given by $R_{n_s}(z) = (1 - zn_s/d)^{-1}$. Applying free decompression, we estimate the $R$-transform of the full matrix to be

$$R_n(z) = \left(1 - z \cdot \frac{n}{n_s} \cdot \frac{n_s}{d}\right)^{-1} = \left(1 - z \cdot \frac{n}{d}\right)^{-1}.$$

This is the $R$-transform of a Marchenko–Pastur law with ratio $\lambda = n/d$, and it approximates the spectral density of a Wishart-distributed matrix with $d$ degrees of freedom and size $n \times n$. This

is precisely what we expected for the full matrix $\mathbf{A}$. In effect, this allows us to infer spectral properties of the $\mathbf{A} = \frac{1}{d}\mathbf{X}\mathbf{X}^\intercal \in \mathbb{R}^{n \times n}$ from the submatrix $\mathbf{A}_{n_s} = \frac{1}{d}\mathbf{X}_{n_s}\mathbf{X}_{n_s}^\intercal \in \mathbb{R}^{n_s \times n_s}$, where the $\mathbf{X}_{n_s} \in \mathbb{R}^{n_s \times d}$ are (much) shorter matrices with (much) smaller aspect ratio $\lambda$ (see Figure 1).

However, for more general classes of matrices, inverting the Stieltjes transform in order to compute its $R$-transform is not analytically tractable. Instead, we will work with the Stieltjes transform directly.

## 3.2 Evolution of the Stieltjes Transform

Our approach can be made precise by appealing to Theorem 1 and operating directly on the $R$ transform. We do this in Proposition 2, which shows that free decompression corresponds to the evolution of a PDE in the Stieltjes transform, given by (1). To do so, for a fixed $R$-transform, we let $R(t, z) = R(ze^t)$, so that $\alpha = n/n_s = e^t$, and we consider the corresponding Stieltjes transform $m(t, z)$ and its inverse $\omega(t, z)$. Then, the following proposition, proved in Appendix E, holds.

**Proposition 2.** *The Stieltjes transform $m(t, z)$ corresponding to R-transforms $R(ze^t)$ satisfies*

$$\frac{\partial m}{\partial t} = -m + m^{-1}\frac{\partial m}{\partial z}. \tag{1}$$

Since (1) is a first-order quasilinear PDE, it can be readily solved using the method of characteristics. This is outlined in Proposition 3, which is proved in Appendix E. A key consequence of this proposition is that we obtain explicit solutions to (1) in terms of the initial data and desired "time" $t$.

**Proposition 3.** *Suppose $m_0$ is analytic in an open domain $\Omega \subset \mathbb{C}$, and $m_0(z) \neq 0$ for all $z \in \Omega$. Consider the initial-value problem (1) with $m(0, z) = m_0(z)$. For $z_0 \in \Omega$, there is a unique solution $m(t, z)$ whose graph can be parametrized by curves $\tau \mapsto (t(\tau), z(\tau), m(\tau))$, $\tau \in \mathbb{R}_{\geq 0}$, where*

$$t(\tau) = \tau, \qquad z(\tau) = z_0 - m_0(z_0)^{-1}(e^\tau - 1), \qquad m(\tau) = m_0(z_0)e^{-\tau}, \tag{2}$$

*valid for $\tau$ such that $z(\tau) \in \Omega$. Furthermore, letting $\Phi(t, z, z_0) := z - z_0 + m(z_0)^{-1}(e^\tau - 1)$, so $\Phi(t(\tau), z(\tau), z_0) = 0$, there exists an inverse function $z_0 = \phi(t, z)$ solving $\Phi(t, z, \phi(t, z)) = 0$ which allows the explicit solution of (1) by $m(t, z) = m_0(\phi(t, z))e^{-t}$.*

Finally, Proposition 4, proved in Appendix F, shows that errors in free decompression can grow, but no more than polynomially fast in the decompression ratio $\frac{n}{n_s}$.

**Proposition 4.** *Let $m$ be a Stieltjes transform of a density $\rho$ such that $\|\rho\|_{L^\infty} < +\infty$, and let $\delta m_{n_s}$ denote the error in an approximation to $m$ starting with a matrix size of $n_s$. For $\delta m_n$, the error under free decompression (1) to a ratio $n/n_s$, there is a constant $\nu > 0$ such that for any $t > 0$,*

$$\|\delta m_n\|_{L^2} \leq (n/n_s)^\nu \|\delta m_{n_s}\|_{L^2}.$$

## 3.3 Heuristic Motivation

While our method is ultimately constructed via Theorem 1, it is illustrative to consider the behavior of the Stieltjes transforms of successive matrices $\mathbf{A}_n$ in our sequence. In what follows, we will proceed informally, roughly following the reasoning outlined in Tao (2017). This heuristic approach is to aid the reader's intuition by viewing the evolution equations in terms of the Schur complement, to motivate the full proof in Appendix E. We let $e_i$ denote the $i$-th column basis vector. Taking two successive matrices in our sequence $\mathbf{A}_n$, we exploit properties of the Schur complement and the Woodbury matrix identity, to obtain the relationship between the trace of their inverses

$$\text{tr}(\mathbf{A}_{n+1} - z\mathbf{I})^{-1} = \text{tr}(\mathbf{A}_n - z\mathbf{I})^{-1} + \frac{e_{n+1}^\intercal(\mathbf{A}_{n+1} - z\mathbf{I})^{-2}e_{n+1}}{e_{n+1}^\intercal(\mathbf{A}_{n+1} - z\mathbf{I})^{-1}e_{n+1}}. \tag{3}$$

Assuming the rows and columns of $\mathbf{A}$ have already been randomly permuted, then the distribution of each row and column of $(\mathbf{A}_{n+1} - z\mathbf{I})^{-1}$ is the same. Therefore, in expectation,

$$\mathbb{E}\left[e_{n+1}^\intercal(\mathbf{A}_{n+1} - z\mathbf{I})^{-1}e_{n+1}\right] = \frac{1}{n+1}\sum_{i=1}^{n+1}\mathbb{E}\left[e_i^\intercal(\mathbf{A}_{n+1} - z\mathbf{I})^{-1}e_i\right] = \frac{1}{n+1}\mathbb{E}\left[\text{tr}(\mathbf{A}_{n+1} - z\mathbf{I})^{-1}\right].$$

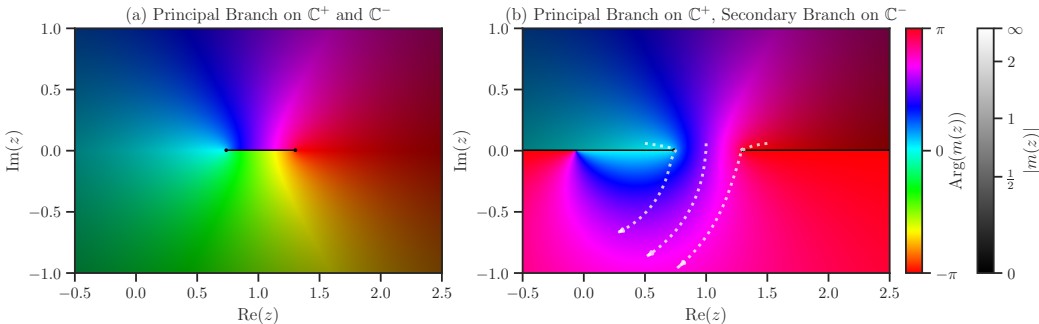

Figure 2: Analytic continuation of the Stieltjes transform of a Marchenko–Pastur distribution (with ratio $\lambda = \frac{1}{50}$). (a) The principal branch contains a branch cut along the support of the spectral density, while (b) the secondary branch is continuous in this region. Curves (white) highlight the evolution of $t \mapsto \phi(t, z)$ over $t \in [0, 4]$, crossing the support of the spectral density.

---

**Algorithm 1:** Pseudocode for Free Decompression

**Input** : random submatrix $\mathbf{A}_{n_s} \in \mathbb{R}^{n_s \times n_s}$ of Hermitian matrix $\mathbf{A} \in \mathbb{R}^{n \times n}$ ;
spectral smoothing function `EigApprox`; gluing function `Glue`;
Stieltjes transform approximation `Stieltjes`, perturbation $\delta$

**Output** : Estimated spectral density values $\{\hat{\rho}(x)\}_{x \in \mathcal{X}}$.

1 Compute the spectrum of $\mathbf{A}_{n_s}$, $\{\lambda_i\}_{i=1}^n$.        *// Step 1: Compute spectrum*
2 $S = \texttt{EigApprox}(\{\lambda_i\}_{i=1}^n)$        *// Step 2: Estimate smoothed spectral density*
3 $G = \texttt{Glue}(S)$        *// Step 3: Estimate glue function*
4 $m(z) = \texttt{Stieltjes}(S, G)$.        *// Step 4: Compute the Stieltjes transform*
5 **foreach** $x \in \mathcal{X}$ **do**
6     Find $z_x \in \mathbb{C}^-$ such that $x + i\delta = z_x - \frac{\frac{n}{n_s} - 1}{m(z_x)}$.        *// Step 5: Solve characteristics*
7 **return** $\left\{ \frac{1}{\pi} \Im\, m(z_x)\, \frac{n_s}{n} \right\}_{x \in \mathcal{X}}$.        *// Step 6: Return the estimated spectral density*

---

The same holds for $\boldsymbol{e}_{n+1}^{\mathsf{T}}(\mathbf{A}_{n+1} - z\mathbf{I})^{-2}\boldsymbol{e}_{n+1}$. This is effectively a Hutchinson trace estimation procedure (Bekas et al., 2007), and we note that the *variance* of the Hutchinson estimators become small for large $n$ as well (Roosta-Khorasani & Ascher, 2015). Assuming this and noting that $\mathrm{tr}(\mathbf{A}_{n+1} - z\mathbf{I})^{-2} = \frac{d}{dz}\mathrm{tr}(\mathbf{A}_{n+1} - z\mathbf{I})^{-1}$, normalizing by $n + 1$ we deduce the difference equation

$$m_{\mathbf{A}_{n+1}}(z) - m_{\mathbf{A}_n}(z) = n^{-1}\left(-m_{\mathbf{A}_n}(z) + m_{\mathbf{A}_n}^{-1}(z)m'_{\mathbf{A}_n}(z)\right) + \mathcal{O}(n^{-2}). \tag{4}$$

For $n$ large, provided $e^t \approx n$ for $t > 0$ so that $\frac{\partial}{\partial t}m(e^t, z) \approx n[m_{\mathbf{A}_{n+1}}(z) - m_{\mathbf{A}_n}(z)]$, (4) is well approximated by the differential equation for the Stieltjes transform $m_{\mathbf{A}}(t, z)$ given by (1).

## 4    Approximating the Stieltjes Transform

In principle, Propositions 2 and 3 suggest a straightforward approach to estimating the Stieltjes transform under free decompression, but these results disguise two main practical challenges:

1. The Stieltjes transform is singular on the support of the spectral density, requiring analytic continuation in a way that does not disrupt the complex transport flow of (1). Without imposing additional properties, numerical analytic continuation is notoriously ill-posed (Trefethen, 2023).
2. The second challenge is accurately approximating the initial Stieltjes transform $m(z)$ from the submatrix eigenvalues. This is a well-studied problem, but as we will see, the nature of the PDE (1) necessitates accomplishing this task to a much higher degree of precision than is typically required.

### 4.1 Analytic Continuation

The Stieltjes transform $m$ is typically defined on the upper half-plane $\mathbb{C}^+ := \{z : \Im(z) > 0\}$ and extended to the lower half-plane $\mathbb{C}^-$ via the Schwarz reflection principle, $m(\bar{z}) = \overline{m(z)}$. This defines the *principal branch*, which exhibits a branch cut along the real axis on $I := \operatorname{supp}(\rho)$. However, the curves $\phi(t, z)$ traced by the PDE dynamics descend from $\mathbb{C}^+$ into $\mathbb{C}^-$, crossing the real axis precisely through $I$, where the principal branch becomes discontinuous. This is formalized in the following proposition.

**Proposition 5.** *Let the curve $t \mapsto \phi(t, z)$ be the solution of characteristic curve given in Proposition 3. Then, for each $z \in \mathbb{C}^+$, there exists $t^* > 0$ such that $\Im\phi(t^*, z) = 0$. In particular, $\Re\phi(t^*, z) \in I$.*

Proposition 5 is proven in Appendix C.1, and this crossing necessitates defining a new branch of the transform—equal to the principal branch in $\mathbb{C}^+$—that extends analytically into $\mathbb{C}^-$ and remains holomorphic across the interior $I^\circ$. The existence of such a continuation is guaranteed by the principle of analytic continuation: since $m$ is holomorphic in $\mathbb{C}^+$ and has no singularities on $I$, it can be extended across the cut into $\mathbb{C}^-$. We refer to this extension as the *secondary branch*.

Figure 2 illustrates this continuation for the Marchenko–Pastur law, showing both branches. Multiple approaches were considered to construct the secondary branch, and are discussed in Appendix D. The most successful approach, and the one considered in experiments to follow, introduces a *gluing function* $G(z)$ defined in (C.5), which compensates for the discontinuity across $I^\circ$. In Appendix C, we show that this function can be accurately approximated by a low-degree rational function (a Padé approximant). When the analytic form of the Stieltjes transform is known, this construction is exact, as shown in Table B.3. In empirical settings—when the density is estimated from the eigenvalues of a submatrix—the same approximation still enables accurate continuation.

In the sequel, all references to the Stieltjes transform $m$ implicitly refer to this secondary, analytically continued branch.

### 4.2 Approximating the Empirical Spectral Density

In light of Proposition 5, free decompression cannot be applied—even in a weak sense—to the empirical Stieltjes transform, $\hat{m}_{\mathbf{A}_n}(z) = \sum_{i=1}^n \frac{1}{\lambda_i - z}$. Instead, we must work with the Stieltjes transform of a smoothed approximation to the empirical spectral distribution. For the synthetic examples we consider, the spectral densities of interest exhibit square-root behavior at the edges of their support, as shown in Table B.1, making Chebyshev polynomials a natural basis for approximation.

However, for real datasets, which may exhibit more complex or irregular spectral behavior, it is often preferable to use the more flexible class of Jacobi polynomials. These introduce two tunable hyperparameters $(\alpha, \beta)$ and recover Chebyshev polynomials when $(\alpha, \beta) = (\frac{1}{2}, \frac{1}{2})$. To fit these polynomials, we found that Galerkin projections of kernel density estimates were often necessary, using Gaussian or Beta kernels. Once the spectral density is approximated in the Jacobi basis, the Stieltjes transform is computed directly using Gauss–Jacobi quadrature (Shen et al., 2011, Section 3.2.2). Details of these procedures and a full algorithmic description of our implementation can be found in Appendix D. Pseudocode for the full free decompression method is provided in Algorithm 1.

## 5 Numerical Examples

We now present numerical examples that demonstrate the utility of our free decompression method. To begin, we consider a number of synthetic examples for random matrices whose spectral densities and their corresponding Stieltjes transforms have known analytic expressions. We then consider two evaluations on real-world large-scale datasets: Facebook Page-Page network (Leskovec & Krevl, 2014; Rozemberczki et al., 2021); and the empirical Neural Tangent Kernel (NTK) corresponding to ResNet50 (He et al., 2016) trained on CIFAR10 (Krizhevsky, 2009). All experiments were conducted on a consumer-grade device with an AMD Ryzen®7 5800X processor, NVIDIA RTX 3080, and 64GB RAM. Hyperparameters used in each experiment are listed in Appendix H.

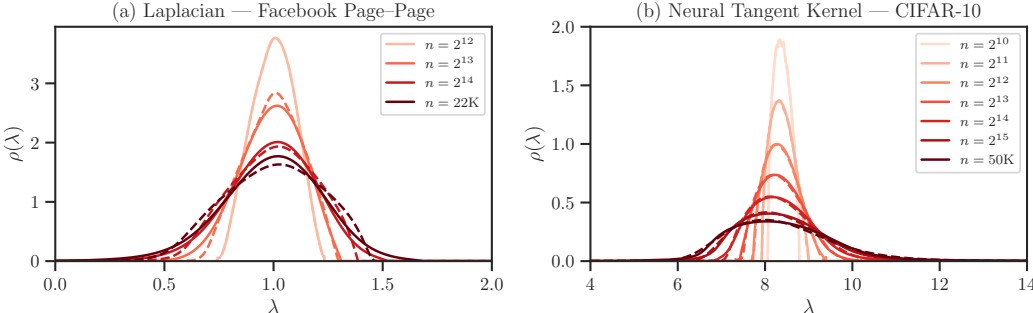

Figure 3: (a) Empirical spectral densities (dashed) of the symmetrically normalized Laplacian matrix of the SNAP Facebook dataset, compared against estimates obtained via free decompression from an initial submatrix of size $2^{12}$ (solid). (b) Empirical spectral densities (dashed) of submatrices of the log-NTK matrix of size $50{,}000$, computed from the CIFAR-10 dataset using a ResNet-50 model, and estimates obtained via free decompression from an initial submatrix of size $2^{10}$ (solid).

**Synthetic Experiments:** We consider five families of distributions that commonly arise as spectral distributions in random matrix theory: the Wigner (1955) semicircle law (free Gaussian); Marčenko & Pastur (1967) (free Poisson); Kesten (1959) and McKay (1981) (free binomial); Wachter (1978) (free Jacobi); and the general free-Meixner family (Saitoh & Yoshida, 2001; Anshelevich, 2008) (free analogs of the classical Meixner (1934) laws). In Table B.1, we list, for each law, its absolutely continuous density $\rho(x)$ supported on $[\lambda_-, \lambda_+]$, and any point masses. Immediately following, Table B.2 summarizes the simplest random-matrix or combinatorial constructions realizing these laws in practice. Figure 1 depicts our free decompression procedure for the Marchenko–Pastur case. We sampled $\mathbf{X} \in \mathbb{R}^{n \times d}$, $n = 1000$, $d = 50{,}000$, with each $X_{ij} \sim \mathcal{N}(0, 1)$. The left panel shows the empirical density of $\mathbf{A} = \frac{1}{d}\mathbf{X}\mathbf{X}^{\mathsf{T}}$ as well as our density approximation using Beta kernels projected onto 50-degree Chebyshev polynomials. The center panel shows a range of densities estimated by free decompression. The right panel compares our freely decompressed approximation of the spectral distribution of $\mathbf{A} \in \mathbb{R}^{n \times n}$, $n = 32{,}000$, with the known exact Marchenko–Pastur law with ratio $\lambda = \frac{32}{50}$. The principal and secondary branches are compared in Figure 2. The remaining models are treated in Appendix B.1, with comparison of the expected log-determinants and distributional distances.

**SNAP Facebook Dataset:** We use the publicly available Facebook Page–Page network from SNAP (Leskovec & Krevl, 2014; Rozemberczki et al., 2021), where nodes represent verified Facebook pages (e.g., politicians, musicians) and edges indicate mutual likes. This graph is undirected with $22{,}470$ nodes and $171{,}002$ edges. We study the eigenvalue distribution of the symmetrically normalized Laplacian matrix of the graph (see e.g., (Mahoney, 2016, p. 16, Definition 4)), which yields a compact spectrum in $[0, 2]$. To avoid artificial spikes caused by degree-1 leaf nodes (which concentrate eigenvalues at $\lambda = 1$), we apply a small Erdős–Rényi perturbation graph $\mathcal{G}(n, p)$ at the connectivity regime where $p = \frac{1}{n}\log(n)$ (Huang & Landon, 2020) to the adjacency matrix of the graph before constructing the Laplacian. Comparison of the empirical spectral densities of randomly sampled submatrices with their free decompression approximations is shown in the left panel of Figure 3.

**Neural Tangent Kernel:** Finally, we consider the spectrum of an empirical NTK derived from a ResNet50 network trained on CIFAR10. The NTK is a well-studied object in machine learning (Jacot et al., 2018; Novak et al., 2022); see Appendix G for background. This example is particularly challenging where naïve approaches fail. The empirical NTK is known to be highly ill-conditioned for well-trained multi-class models (Ameli et al., 2025). When the NTK is constructed in blockwise fashion, classification models with $c$ classes often exhibit a low-rank structure, with approximately $1/c$ of the eigenvalues being extremely small. To address this, we project the NTK matrix onto its non-null eigenspace, removing the near-zero eigenvalues, and then reconstruct a Hermitian matrix in the reduced space. Subsamples are drawn from this reduced matrix. Specifically, we took $\hat{\mathbf{A}}$ to be the full $55{,}560 \times 55{,}560$ matrix (corresponding to 5556 images), and reduced this to a $50{,}000 \times 50{,}000$ matrix $\mathbf{A}$ after removing the null component. Submatrices of dimension $1024, 2048, 4096, 8192, 16{,}382,$

Table 2: Comparison of process time for direct computation of the spectral density versus free decompression (FD) on the NTK dataset. In the FD column, the first term is the time to compute eigenvalues of the initial $2^{10}$-sized submatrix (common to all rows) and the second term is the FD step. Metrics (columns 4–8) compare the true spectral density with the FD approximation, and are reported as distributional distances (total variation, Jensen–Shannon, Kolmogorov–Smirnov) and relative errors of the first and second moments, all multiplied by 100 to give percentages. Values are the mean over 20 randomly sampled initial submatrices with standard deviations in parentheses.

| Size | Process Time (sec) | | Distributional Distances ($\times 100$) | | | Moments Rel. Error ($\times 100$) | |
| --- | --- | --- | --- | --- | --- | --- | --- |
| $n_s$ | Direct | FD (ours) | TV | JS | KS | $\Delta\mu_1/\mu_1$ | $\Delta\mu_2/\mu_2$ |
| $2^{10}$ | 3.6 | $3.6 + 0.0$ | 0.00% ($\pm 0.00$) | 0.00% ($\pm 0.00$) | 0.00% ($\pm 0.00$) | 0.00% ($\pm 0.00$) | 0.00% ($\pm 0.00$) |
| $2^{11}$ | 10.2 | $3.6 + 0.6$ | 1.72% ($\pm 0.37$) | 7.60% ($\pm 1.99$) | 0.48% ($\pm 0.10$) | 0.05% ($\pm 0.03$) | 0.09% ($\pm 0.04$) |
| $2^{12}$ | 50.9 | $3.6 + 0.6$ | 2.06% ($\pm 0.35$) | 4.67% ($\pm 0.67$) | 0.70% ($\pm 0.12$) | 0.01% ($\pm 0.01$) | 0.02% ($\pm 0.02$) |
| $2^{13}$ | 358.9 | $3.6 + 0.6$ | 3.24% ($\pm 0.56$) | 6.30% ($\pm 0.51$) | 1.18% ($\pm 0.22$) | 0.01% ($\pm 0.00$) | 0.02% ($\pm 0.01$) |
| $2^{14}$ | 2820.2 | $3.6 + 0.7$ | 4.33% ($\pm 0.87$) | 7.55% ($\pm 0.80$) | 1.76% ($\pm 0.48$) | 0.01% ($\pm 0.01$) | 0.03% ($\pm 0.01$) |
| $2^{15}$ | 20451.2 | $3.6 + 0.8$ | 5.16% ($\pm 1.18$) | 7.96% ($\pm 1.11$) | 2.51% ($\pm 0.84$) | 0.02% ($\pm 0.01$) | 0.05% ($\pm 0.02$) |
| 50K | 67331.1 | $3.6 + 0.8$ | 5.94% ($\pm 1.48$) | 8.33% ($\pm 1.41$) | 3.02% ($\pm 1.14$) | 0.17% ($\pm 0.03$) | 0.49% ($\pm 0.06$) |

and 32,768 were then sampled, and their empirical spectral distributions are shown as dashed lines in the right panel of Figure 3. Free decompression approximations based on the Stieltjes transform of the $1024 \times 1024$ submatrix (see Figure G.1) are shown as solid lines. Table 2 reports the corresponding process times and errors in total variation (TV) distance, Jensen–Shannon (JS) distance, Kolmogorov–Smirnov (KS) distance, and the relative error in the first and second moments.

## 6   Conclusion

In modern settings, it is becoming increasingly common that the spectrum information of a matrix is required, even when the matrix is too large to fit into memory, or is otherwise unavailable. These matrices comprise the category of *impalpable matrices*, offering unique challenges to numerical linear algebra practitioners. We have outlined *free decompression*, a novel method of approximating the spectral density of a large impalpable Hermitian matrix from the spectrum of a randomly sampled submatrix. Our initial implementation of this method shows promise as a tool to extract accurate information about the spectrum of extremely large matrices that would otherwise be unavailable. We hope that this work motivates other researchers to further develop these and similar tools to further unlock information in impalpable matrices that is otherwise out of reach.

**Limitations.** While our method imposes few restrictions on the large matrix of interest, the quality of approximation depends strongly on asymptotic estimates of the spectral density of $\mathbf{A}_n$. While promising for future research, performing free decompression accurately remains highly challenging, with Proposition 4 highlighting the need for accurate spectral density estimation and analytic continuation. Without a general approach to estimate and operate on the Stieltjes transform on log-scale, that is, estimating $z \mapsto m(e^z)$, highly ill-conditioned matrices still remain out of reach. At present, the implementation of our method is sensitive to the quality of approximation of the gluing function; and, while our approach works flawlessly in simple scenarios, it becomes more delicate with real data. Nevertheless, we stress that despite these limitations, no competing alternative exists for estimating the spectral density of impalpable matrices.

## Acknowledgments and Disclosure of Funding

LH is supported by the Australian Research Council through a Discovery Early Career Researcher Award (DE240100144). MWM acknowledges partial support from DARPA, DOE, NSF, and ONR. This work was supported in part by the U.S. Department of Energy, Office of Science, Office of Advanced Scientific Computing Research's Applied Mathematics Competitive Portfolios program under Contract No. AC02-05CH11231.

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

# Appendices

## Contents

## Appendix A    Nomenclature

We use boldface lowercase letters for vectors, boldface upper case letters for matrices, and normal face letters for scalars, including the components of vectors and matrices. Table A.1 summarizes the main symbols and notations used throughout the paper, organized by context.

## Appendix B    Benchmark Matrix Ensembles

We evaluate our method on a suite of benchmark matrix ensembles with known limiting spectral distributions. Appendix B.1 reviews the underlying matrix models and associated laws, while Appendix B.2 presents decompression results on these closed-form distributions.

### B.1    Overview of Random Matrix Ensembles

We review several canonical spectral laws arising from random matrix and combinatorial models, which serve as ground-truth distributions in our benchmarks. Concretely, we consider the Wigner (1955) semicircle law (free Gaussian), Marčenko & Pastur (1967) (free Poisson), Kesten (1959) and McKay (1981) (free binomial), Wachter (1978) (free Jacobi), and the general free-Meixner family (Saitoh & Yoshida, 2001; Anshelevich, 2008) (free analogs of the classical Meixner (1934) laws).

---

[2]This definition differs by a factor of $-1/\pi$ from the more common definition $\frac{1}{\pi}$ p. v. $\int \frac{\rho(y)}{x-y}\,dy$.

Table A.1: Common notations used throughout the manuscript.

| Context | Symbol | Description |
|---|---|---|
| Free Probability | $\rho$ | Absolutely continuous empirical spectral density, $\mathbb{R} \to \mathbb{R}^+$ |
| | $I$ | Compact support interval of density $\rho$, $I = [\lambda_-, \lambda_+] \subsetneq \mathbb{R}$ |
| | $m$ | Stieltjes transform of density, $m(z) = \int_{\mathbb{R}} \frac{\rho(y)}{y-z} \, dy$, $\mathbb{C} \setminus I \to \mathbb{C}$ |
| | $\mathcal{H}[\rho]$ | Hilbert transform[2] of density, $\mathcal{H}[\rho](x) = \mathrm{p.\,v.} \int_{\mathbb{R}} \frac{\rho(y)}{y-x} \, dy$, $\mathbb{R} \to \mathbb{R}$ |
| | $R$ | Voiculescu's R-transform, $R(z) = \omega(-z) - \frac{1}{z}$ with $\omega(m(z)) = z$ |
| Free Decomposition | $n$ | Size of the target (large) matrix |
| | $n_s$ | Size of the sampled (small) sub-matrix, $n_s < n$ |
| | $t$ | Decompression scale, $t = \log\left(\frac{n}{n_s}\right)$ |
| | $\rho_0, m_0$ | Density and Stieltjes transform at $t = 0$ |
| Notation | $\mathbb{C}^\pm$ | Upper (lower) half complex plane |
| | $\Re, \Im$ | Real and imaginary part of complex variable |
| | $\mathrm{p.\,v.}$ | Cauchy principal value |

In Table B.1, we list, for each law, its absolutely continuous density $\rho(x)$ supported on $I = [\lambda_-, \lambda_+]$, and any point masses. Immediately following, Table B.2 summarizes the simplest random-matrix or combinatorial constructions realizing these laws in practice.

Table B.1: Spectral laws used in our analytic benchmarks, along with their free-probability analogs, absolutely continuous density functions $\rho(x)$, compact support intervals $[\lambda_-, \lambda_+]$, and any point-mass atoms.

| Distribution | Free Corresp. | Abs. Cont. Density $\rho(x)$ | Support $\lambda_\pm$ | Number of Atoms |
|---|---|---|---|---|
| Wigner semicircle | Free Gaussian | $\dfrac{2\sqrt{r^2 - x^2}}{\pi r^2}$ | $\pm r$ | None |
| Marchenko–Pastur | Free Poisson | $\dfrac{\sqrt{(\lambda_+ - x)(x - \lambda_-)}}{2\pi \lambda x}$ | $(1 \pm \sqrt{\lambda})^2$ | $(1 - \frac{1}{\lambda})\delta(x)$ if $\lambda > 1$ |
| Kesten–McKay | Free Binomial | $\dfrac{d\sqrt{4(d-1) - x^2}}{2\pi(d^2 - x^2)}$ | $\pm 2\sqrt{d-1}$ | None |
| Wachter | Free Jacobi | $\dfrac{(a+b)\sqrt{(\lambda_+ - x)(x - \lambda_-)}}{2\pi x(1-x)}$ | $\left(\dfrac{\sqrt{b} \pm \sqrt{a(a+b-1)}}{a+b}\right)^2$ | $x = 0, 1$ |
| Meixner | Free Meixner | $\dfrac{c\sqrt{4b - (x-a)^2}}{2\pi((1-c)x^2 + acx + bc^2)}$ | $a \pm 2\sqrt{b}$ | At most two |

All of the above distributions arise—up to affine rescaling—from the free Meixner class of measures (Anshelevich, 2007), whose orthogonal-polynomial recursion coefficients stabilize after the first level. Equivalently, the associated Jacobi matrix takes the bordered–Toeplitz form

$$\mathbf{J} = \begin{bmatrix} \alpha_0 & \beta_0 & & \\ \hline \beta_0 & \alpha_1 & \beta_1 & \\ & \beta_1 & \alpha_1 & \beta_1 \\ & & \ddots & \ddots & \ddots \end{bmatrix}, \quad \alpha_n = \alpha_1, \ \beta_n = \beta_1 \quad (n \geq 1).$$

See Dubbs & Edelman (2015, Table 2) for the specific $(\alpha_1, \beta_1)$ giving each law. In particular, setting $\alpha_0 = 0$, $\beta_0 = 1$ yields the standard zero-mean, unit-variance normalization.

Table B.2: Empirical realizations of the spectral laws listed in Table B.1. The matrix $\mathbf{X}$ is assumed to have i.i.d. standard normal entries.

| Distribution | Matrix or Combinatorial Model | Parameters |
|---|---|---|
| Wigner semicircle | Gaussian orthogonal ensemble $\frac{1}{\sqrt{2}}(\mathbf{X} + \mathbf{X}^{\mathsf{T}})$ | $r = 2\sqrt{n}$ |
| | Adjacency matrix of Erdős–Rényi graph $G(n, p)$ | $pn = \mathcal{O}(\log(n))$ |
| Marchenko–Pastur | Sample covariance (Wishart) $\frac{1}{d}\mathbf{X}\mathbf{X}^{\mathsf{T}}$, $\mathbf{X} \in \mathbb{R}^{n \times d}$ | $\lambda = \frac{n}{d}$ |
| Kesten–McKay | Haar–orthogonal Hermitian sum $\sum_{i=1}^{k}(\mathbf{O}_i + \mathbf{O}_i^{\mathsf{T}})$ | $d = 2k$ |
| | Projection model $d\,\mathbf{P}\mathbf{O}\mathbf{D}\mathbf{O}^{\mathsf{T}}\mathbf{P}$ (Longoria & Mingo, 2023) | $d \geq 2$ |
| | Adjacency matrix of a random $d$-regular graph | $d \geq 2$ |
| Wachter | Generalized eigenvalues of $(\mathbf{S}_1, \mathbf{S}_1 + \mathbf{S}_2)$, $\mathbf{S}_i = \frac{1}{d_i}\mathbf{X}_i\mathbf{X}_i^{\mathsf{T}}$ | $a = \frac{d_1}{n}, b = \frac{d_2}{n}$ |
| | Arises in MANOVA problems | |
| Meixner | Bordered Toeplitz tridiagonal with Jacobi coefficients $\alpha_1, \beta_1$ | $a = \alpha_1, b = \beta_1 - 1$ |
| | Block–Gaussian ensembles (Lenczewski, 2015) | |

Since the Jacobi parameters in the above become constant beyond the second level, the continued-fraction expansion of the Stieltjes transform

$$m(z) = \cfrac{1}{z - \alpha_0 - \cfrac{\beta_0^2}{z - \alpha_1 - \cfrac{\beta_1^2}{z - \alpha_1 - \cfrac{\beta_1^2}{\ddots}}}},$$

becomes eventually periodic, exhibiting a repeating tail that we encode by

$$m(z) = \frac{1}{z - \alpha_0 - \beta_0^2 T}, \quad \text{where} \quad T = \frac{1}{z - \alpha_1 - \beta_1^2 T}.$$

Eliminating $T$ yields the quadratic equation

$$Q(z)m(z)^2 - P(z)m(z) + 1 = 0,$$

where

$$P(z) = 2(z - \alpha_0) - \frac{\beta_0^2}{\beta_1^2}(z - \alpha_1),$$

$$Q(z) = (z - \alpha_0)^2 - \frac{\beta_0^2}{\beta_1^2}(z - \alpha_0)(z - \alpha_1) + \frac{\beta_0^4}{\beta_1^2}.$$

Note that $Q(z)$ is at most quadratic (degenerating to linear when $\beta_0^2 = \beta_1^2$), and $P$ is at most linear. Solving for $m(z)$ yields

$$m(z) = \frac{P(z) + \sqrt{P(z)^2 - 4Q(z)}}{2Q(z)}, \tag{B.1}$$

where the branch of the square root is chosen such that $m(z)$ defines a Herglotz map $\mathbb{C}^+ \to \mathbb{C}^+$. The corresponding Hilbert transform $\mathcal{H}[\rho](x) = \Re\, m(x + i0)$ inside the support interval $I = [\lambda_-, \lambda_+]$ is thus the rational function

$$\mathcal{H}[\rho](x) = \frac{P(x)}{2Q(x)}, \quad x \in I. \tag{B.2}$$

Table B.3 lists these polynomials $P, Q$ and the corresponding free cumulant $R$-transforms for every distribution in Table B.1.

Table B.3: Stieltjes, Hilbert, and $R$-transforms for the spectral laws listed in Table B.1. The Stieltjes and Hilbert transforms are constructed from the polynomials $P(z)$ and $Q(z)$ according to (B.1) and (B.2).

| | Stieltjes and Hilbert Transforms | | |
| Distribution | $P(z)$ | $Q(z)$ | $R$-Transform |
| --- | --- | --- | --- |
| Wigner semicircle | $-z$ | $\dfrac{r^2}{4}$ | $\dfrac{r^2}{4}z$ |
| Marchenko–Pastur | $1 - \lambda - z$ | $\lambda z$ | $\dfrac{1}{1 - \lambda z}$ |
| Kesten–McKay | $\dfrac{(2-d)z}{d-1}$ | $\dfrac{d^2 - z^2}{d-1}$ | $\dfrac{-d + d\sqrt{1 + 4z^2}}{2z}$ |
| Wachter | $\dfrac{a - 1 - (a+b-2)z}{a+b-1}$ | $\dfrac{z(1-z)}{a+b-1}$ | $\dfrac{-(a+b) + z + \sqrt{(a+b)^2 + 2(a-b)z + z^2}}{2z}$ |
| Meixner | $(c-2)z - ac$ | $(1-c)z^2 + acz + bc^2$ | $\left(\dfrac{c}{1-c}\right)\dfrac{1 - az + \sqrt{(1-az)^2 - 4b(1-c)z^2}}{2z}$ |

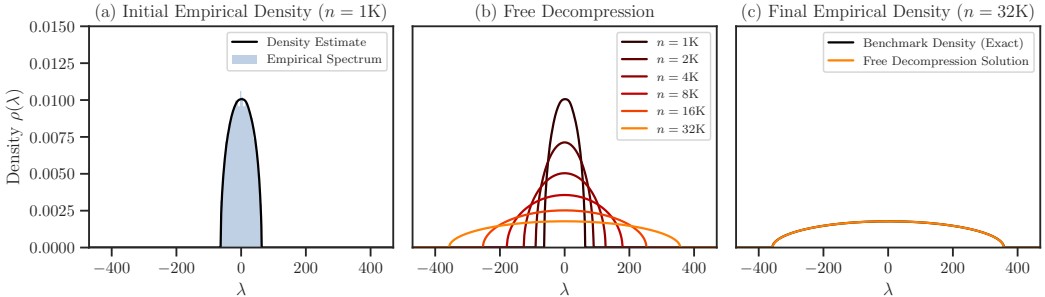

Figure B.1: (a) Estimation of the Wigner semi-circle law for $\mathbf{A} = (\mathbf{X} + \mathbf{X}^\mathsf{T})/\sqrt{2}$, where $\mathbf{X}$ is a $n \times n$ matrix with standard normal entries for $n = 1000$; (b) approximation of densities for various values of $n$ via free decompression (Algorithm 1); and (c) comparison of our decompressed approximation of $n = 32{,}000$ with the known limiting analytic density.

## B.2 Free Decompression of Closed-Form Benchmark Distributions

In addition to the Marchenko–Pastur case discussed earlier (Figure 1), free decompression was tested on a matrices with Wigner, Kesten–McKay, Wachter, and Meixner limiting spectral distributions, as shown in Figures B.1 to B.4, respectively. To identify the Meixner distribution under free decompression, we make use of the $R$-transform. Let $R_{a,b,c}(z)$ denote the $R$-transform for the Meixner distribution with parameters $a, b, c$ as given in Table B.3. A few calculations show that for any $\alpha > 0$, it holds $R_{a,b,c}(\alpha z) = R_{a(\alpha),b(\alpha),c(\alpha)}(z)$ where

$$a(\alpha) = a\alpha, \qquad b(\alpha) = \frac{b\alpha^2(1 - c(\alpha))}{1 - c}, \qquad c(\alpha) = \frac{c}{c + \alpha(1 - c)}.$$

Consequently, performing free decompression on a Meixner distribution with parameters $a, b, c$ by a factor $\alpha = n/n_s$ will yield a Meixner distribution with parameters $a(n/n_s)$, $b(n/n_s)$ and $c(n/n_s)$.

For the Marchenko–Pastur density we also compared the spectral samples of our free decompression estimates with those taken from the analytic density, in order to measure sensitivity to the initial sample that was decompressed. We used three measures of accuracy: total variation, Jensen–Shannon distance, and log-determinants. To this end, free decompression was performed on ten different matrices of size $n = 1000$, each with $d = 50{,}000$ degrees of freedom, to obtain approximations of the spectral density corresponding to 32,000 dimensional expansions. Eigenvalues were then sampled from the corresponding distributions via quasi-Monte Carlo, and compared to samples from the analytic density with aspect ratio $\lambda = \frac{32}{50}$. The average distance in total variation was found to be $0.2\%$, while the average Jensen–Shannon distance was $1.867\%$. The mean log-determinant was

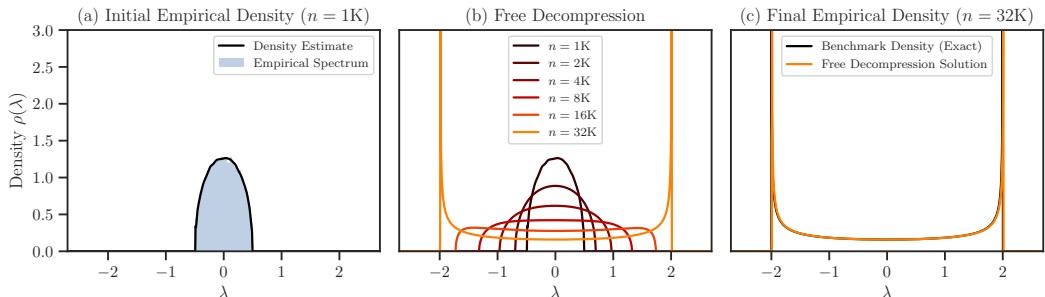

Figure B.2: Estimation of the Kesten–McKay law from the model $\mathbf{A} = \sum_{i=1}^{k}(\mathbf{O}_i + \mathbf{O}_i^{\mathsf{T}})$ (Longoria & Mingo, 2023, Section 5), with $d = 2k = 2$ and size $n = 32{,}000$, where $\mathbf{O}_i$ are Haar–orthogonal matrices generated via Mezzadri (2007). (a) Spectral density from a subsample of size $n_s = 1000$. (b) Densities for various $n$ via free decompression (Algorithm 1). (c) Comparison of decompressed and exact density at $n = 32{,}000$.

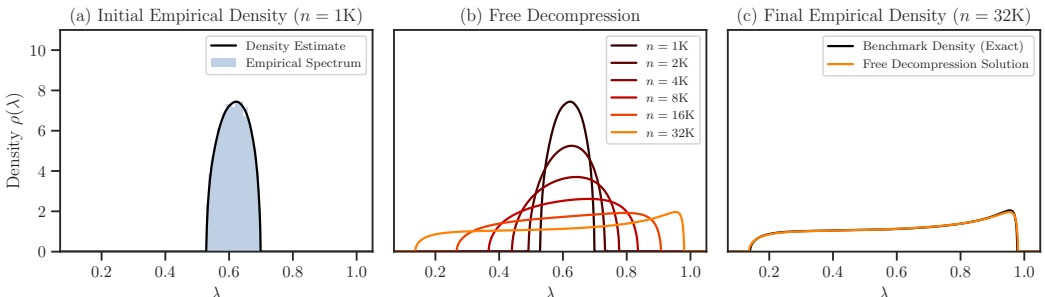

Figure B.3: (a) Estimation of the Wachter distribution with ratios $a = 80$, $b = 50$, from eigenvalues of Wishart matrices with $d_1 = 80{,}000$ and $d_2 = 50{,}000$ degrees of freedom, respectively, and size $n = 1000$; (b) approximation of densities for various values of $n$ via free decompression (Algorithm 1); and (c) comparison of our decompressed approximation of $n = 32{,}000$ with the known limiting analytic density for $a = \frac{80}{32}$, $b = \frac{50}{32}$.

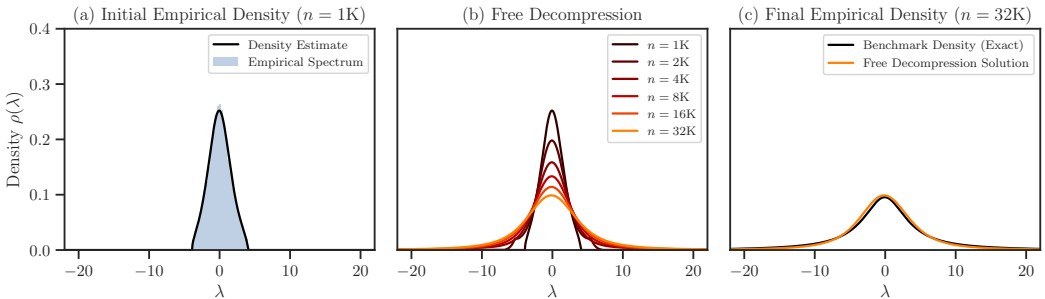

Figure B.4: (a) Estimation of the Meixner distribution with $a = 0.1$, $b = 4$, $c = 0.6$ from $n = 1000$ quasi-Monte Carlo samples; (b) approximation of densities for various values of $n$ via free decompression (Algorithm 1); and (c) comparison of our decompressed approximation of $n = 32{,}000$ with the known analytic density given by the Meixner distribution with $a \approx 0.566$, $b \approx 253$, $c \approx 0.210$.

computed to be $-13514.88$ with a standard deviation of $1.503$, versus the baseline of $-13538.94$. This corresponds to an absolute relative error of $1.78\%$ with standard error of $0.01\%$.

## Appendix C    Considerations for Solving the PDE Using Characteristics

Our task is to compute $m(t, z)$ for a grid of points $z \in \mathbb{C}^+$ and times $t \in [0, \tau]$, given the initial condition $m(0, z) = m_0(z)$. By Proposition 3, one has the explicit representation $m(t, z) = m_0(z_0)e^{-t}$ where the "initial label" $z_0$ satisfies the characteristic relation

$$z = z_0 - \frac{e^t - 1}{m_0(z_0)}. \tag{C.1}$$

Equivalently, each characteristic is a straight line emanating from $z_0$ with direction vector $-m_0(z_0)^{-1}$.

To evaluate $m(t, z)$ at a given $(t, z)$, one must solve the nonlinear equation (C.1) for the unknown initial point $z_0 = \phi(t, z)$. Although this root-finding is numerically tractable, it raises two interlocking issues. First, the map $t \mapsto \phi(t, z)$ departs the upper half-plane and crosses the real axis *through* the spectral support $I := \text{supp}(\rho_0)$, where $m_0$ is discontinuous. Second, evaluating $m_0$ on the lower half-plane requires a holomorphic extension of $m_0$ across the cut $I$.

In the remainder of this section, we address these points in two stages. In Appendix C.1, we prove that every trajectory $\phi(t, z)$ indeed intersects $\mathbb{R}$ precisely on $I$. In Appendix C.2, we construct a second-sheet continuation of $m_0$, holomorphic across $I$, via an additive Riemann-Hilbert (*gluing*) ansatz.

### C.1    Challenge: Crossing the Cut

Consider a fixed target point $z \in \mathbb{C}^+$. We seek to determine the initial points $z_0 = \phi(t, z)$ that, under the flow of the characteristic equation (C.1), pass through $z$ at time $t$. Define the curve $\mathcal{C} : t \mapsto z_0$ by $z_0(t) = \phi(t, z)$, given implicitly as the solution of (C.1) for fixed $z$. This curve traces the locus of initial conditions $z_0$ whose corresponding characteristic trajectories intersect the point $z$ at time $t$.

Unlike the characteristic curves themselves—which are straight lines—this curve $z_0(t)$ is highly nonlinear, beginning at $z_0(0) = z$. In the following propositions, we show that this curve descends from its initial point $z \in \mathbb{C}^+$, eventually crossing into the lower half-plane $\mathbb{C}^-$, and in fact intersects the real axis precisely on the support of $\rho$.

**Proposition C.1** (Characteristic curve exiting $\mathbb{C}^+$). *Let $\rho_0$ be an absolutely continuous probability density (with respect to Lebesgue measure) supported on a compact interval $I \subset \mathbb{R}$. Assume there exist a compact subinterval $I' \subset \text{int } I$ and a constant $c_0 > 0$ such that $\rho_0(x) \geq c_0$ for all $x \in I'$. Let $m_0(z)$ denote the Stieltjes transform of $\rho_0$, which defines a Herglotz map $\mathbb{C}^+ \to \mathbb{C}^+$. For each $z \in \mathbb{C}^+$ and $t > 0$, define $\phi(t) := \phi(t, z) \in \mathbb{C}$ implicitly by*

$$z = \phi(t) - \frac{e^t - 1}{m_0(\phi(t))}. \tag{C.2}$$

*Then there exists a finite time $t^* = t^*(z) > 0$ such that $\Im\phi(t) > 0$ for $0 \leq t < t^*$ and $\Im\phi(t^*) = 0$.*

**Proof.** Let $a(t) = e^t - 1$ and define $w(z) = -1/m_0(z)$, so that $w : \mathbb{C}^+ \to \mathbb{C}^+$. Fix $z = x + iy \in \mathbb{C}^+$. Write $\phi(t) = u(t) + iv(t)$ and $w(\phi(t)) = \alpha(t) + i\beta(t)$ with $\beta(t) > 0$.

First, we show that when the density $\rho_0$ has no atoms, $\phi(t)$ does not asymptote to the edges of $I$ as $t \to \infty$. Taking real parts in (C.2) gives

$$x = u(t) + a(t)\alpha(t).$$

If $u(t) \to u^* \in \partial I$ and $v(t) \to 0^+$, then by Plemelj–Sokhotski $m_0(u^* + i0) \in \mathbb{R}$ is finite (no atom at $u^*$), hence $\alpha(t) \to \alpha^* := \Re w(u^* + i0) = -1/m_0(u^* + i0) \neq 0$, which forces $x = u(t) + a(t)\alpha(t) \to u^* + a(t)\alpha^*$, a contradiction since $a(t) \to \infty$. Thus the trajectory cannot converge to an endpoint while $v(t) > 0$. This rules out the pathological possibility that the trajectory merely approaches the boundary without ever entering the support. Consequently, we may assume the curve eventually moves inside the interior of $I$, where the following estimates apply.

Next, we show $\beta$ is bounded below. By the Plemelj boundary values, for $u \in I$ we have

$$\lim_{v \to 0^+} \Im w(u + iv) = \frac{\pi \rho_0(u)}{|m_0(u + i0)|^2}.$$

Since $\rho_0 \geq c_0 > 0$ on $I'$ and $|m_0(u + i0)|$ is continuous on $I'$, there exist $v_0 > 0$ and $c > 0$ such that $\Im w(u + iv) \geq c$ for all $u \in I'$ and $0 < v \leq v_0$. Hence, once $u(t) \in I'$ with $v(t) \leq v_0$, we have $\beta(t) \geq c$.

Finally, taking imaginary parts in (C.2) gives

$$v(t) = y - a(t)\,\beta(t).$$

As $t \to \infty$, we have $a(t) \to \infty$ while $\beta(t)$ remains bounded below by $c$ whenever $u(t) \in I'$ and $0 < v(t) \leq v_0$. Thus, $v(t) \to -\infty$, so by continuity there exists a first time $t^*$ where $v(t^*) = 0$. $\quad\square$

**Proposition C.2** (Crossing on the support). *Suppose the hypotheses of Proposition C.1 hold, and define the curve $t \mapsto \phi(t, z) \in \mathbb{C}$ implicitly by (C.2) for a fixed $z \in \mathbb{C}^+$. Let $t^* > 0$ be the first time such that $\Im \phi(t^*) = 0$, whose existence is guaranteed by Proposition C.1. Then the real part of the crossing point lies in the support of the density: $\Re \phi(t^*) \in I$.*

**Proof.** Let $x^* := \phi(t^*)$ be the real-valued point where the curve $\phi$ intersects the real axis. Suppose, for contradiction, that $x^* \notin I$. Since $I$ is compact, there exists an open interval $J \subset \mathbb{R} \setminus I$ containing $x^*$. On such an interval $J$, the Stieltjes transform $m_0(x)$ admits real boundary values. Indeed, by the Plemelj–Sokhotski formula,

$$\lim_{\epsilon \to 0^+} m_0(x + i\epsilon) = \mathcal{H}[\rho_0](x) + i\pi\rho_0(x), \quad x \in \mathbb{R}, \tag{C.3}$$

and since $\rho_0(x) = 0$ for all $x \in J$, the imaginary part vanishes and $m_0(x)$ is real-valued on $J$. In particular, $m_0(x^*) \in \mathbb{R}$. Substituting $\phi(t^*) = x^*$ into (C.2), we find that the right-hand side of the equation is real, so $z \in \mathbb{R}$, contradicting the assumption that $z \in \mathbb{C}^+$. Therefore, the assumption $x^* \notin I$ is false, and we conclude $x^* \in I$. $\quad\square$

Since the trajectory $\mathcal{C}$ enters the lower half-plane, we must evaluate the initial field $m_0$ both across the real line at points on $\mathrm{supp}(\rho_0)$, and inside $\mathbb{C}^-$. However, by definition, the Stieltjes transform

$$m_0(z) = \int_I \frac{\rho_0(x)}{x - z}\,\mathrm{d}x,$$

is only defined for $\Im(z) > 0$, and exhibits a jump discontinuity across the real axis precisely on $\mathrm{supp}(\rho_0)$. This creates an analytic obstruction: to evaluate $m_0(\phi(t, z))$ for arbitrary $t$, we require a holomorphic continuation of $m_0$ to the lower half-plane that is smooth across the cut $I$. This continuation is constructed in the next section.

## C.2 Holomorphic Continuation of Stieltjes Transform

Let $m^+(z)$ denote the Stieltjes transform corresponding to a density $\rho$ supported on a compact interval $I \subset \mathbb{R}$ (such as the field $m_0(z)$ and the density $\rho_0$ in our case). By definition, this function is a Herglotz map $\mathbb{C}^+ \to \mathbb{C}^+$, holomorphic on $\mathbb{C}^+$, and admits boundary values on $\mathbb{R}$ via the Sokhotski–Plemelj theorem:

$$\lim_{\epsilon \to 0^+} m^+(x + i\epsilon) = \mathcal{H}[\rho](x) + i\pi\rho(x).$$

A natural extension of $m^+$ to $\mathbb{C}^-$ is the Schwarz reflection

$$m^+(z) = \overline{m^+(\bar{z})}, \quad \Im(z) < 0.$$

This extension is holomorphic on $\mathbb{R} \setminus I$, but has a discontinuity across $I$. Specifically, the discontinuity arises from a jump in the imaginary part:

$$\lim_{\epsilon \to 0^+} \Im(m^+(x + i\epsilon)) - \Im(m^+(x - i\epsilon)) = 2\pi\rho(x).$$

Thus, $\Im(m^+)$ is discontinuous on $I$, while it vanishes and remains continuous on $\mathbb{R} \setminus I$. We also note that the real part, $\Re(m^+) = \mathcal{H}[\rho](x)$, is continuous across the entire real axis.

Our goal is to construct a second analytic continuation of $m^+$, denoted $m^-$, that is holomorphic across the interior of the cut $I$. That is, $m^-$ agrees with $m^+$ on $\mathbb{C}^+$ and continues analytically to $\mathbb{C}^-$

through $I$, at the expense of developing a discontinuity across $\mathbb{R} \setminus I$. This setup forms an additive Riemann–Hilbert problem: find two branches $m^+$ and $m^-$ such that

$$\lim_{\epsilon \to 0^+} m^+(x + i\epsilon) = \lim_{\epsilon \to 0^+} m^-(x - i\epsilon), \quad x \in I.$$

We propose the following structure:

$$m^-(z) := \begin{cases} m^+(z), & \Im(z) > 0, \\ -m^+(z) + G(z), & \Im(z) < 0, \end{cases} \tag{C.4}$$

where $G(z)$ is a *glue function* chosen to cancel the jump across $I$. In particular, we require that

$$\begin{aligned} \Im(G(x + i0)) &= 0 & x \in I, \\ \Re(G(x + i0)) &= 2\mathcal{H}[\rho](x), & x \in I. \end{aligned} \tag{C.5}$$

The first condition in the above ensures the continuity of the imaginary part on the whole of $\mathbb{R}$:

$$\lim_{\epsilon \to 0^+} \Im(m^-(x + i\epsilon)) - \Im(m^-(x - i\epsilon)) = \pi\rho(x) - \pi\rho(x) = 0,$$

and the second condition guarantees continuity of the real part across $I$:

$$\lim_{\epsilon \to 0^+} \Re(m^-(x + i\epsilon)) - \Re(m^-(x - i\epsilon)) = 0,$$

while allowing discontinuities on $\mathbb{R} \setminus I$, which are harmless for our purposes. In this way, $m^+$ and $m^-$ form two complementary Riemann sheets: $m^+$ is holomorphic in $\mathbb{C}^+ \cup (\mathbb{R} \setminus I)$ but discontinuous across $I$, while $m^-$ is holomorphic in $\mathbb{C}^- \cup I$, with a branch cut on $\mathbb{R} \setminus I$.

To characterize $G$, note that $G(z) = m^-(z) + m^+(z)$, and it must be analytic in $\mathbb{C}^-$. On the real axis inside the support, this glue function satisfies $\Re G(x) = 2\mathcal{H}[\rho](x)$, and thus inherits key structural properties from the Hilbert transform. In order to design an efficient functional form for $G$, we first study the behavior of $\mathcal{H}[\rho]$ itself in Proposition C.3, which guides the minimal rational approximation ansatz in the next step.

**Proposition C.3** (Properties of the Hilbert transform). *Let $\rho$ be a bounded, piecewise-$C^1$ density supported on a connected compact interval $I = [\lambda_-, \lambda_+] \subset \mathbb{R}$. Define the Hilbert transform*

$$\mathcal{H}[\rho](x) = \mathrm{p.\,v.} \int_{\lambda_-}^{\lambda_+} \frac{\rho(t)}{t - x} \, \mathrm{d}t,$$

*where* p.v. *denotes the Cauchy principal value. Then $\mathcal{H}[\rho]$ and its derivative with respect to $x$ satisfy:*

1. *For $x \in (-\infty, \lambda_-)$: $\mathcal{H}[\rho](x) > 0$ and $\mathcal{H}[\rho]'(x) > 0$.*

2. *For $x \in [\lambda_-, \lambda_+]$: the equation $\mathcal{H}[\rho](x) = 0$ has an odd (hence at least one) number of solutions $x^* \in (\lambda_-, \lambda_+)$.*

3. *For $x \in (\lambda_+, \infty)$: $\mathcal{H}[\rho](x) < 0$ and $\mathcal{H}[\rho]'(x) > 0$.*

4. *As $x \to \pm\infty$, $\mathcal{H}[\rho](x) \sim -1/x$.*

**Proof.** *Step 1. Outside the support.* If $x \notin [\lambda_-, \lambda_+]$, differentiation under the integral sign is valid:

$$\mathcal{H}[\rho]'(x) = \int_{\lambda_-}^{\lambda_+} \frac{\rho(t)}{(t - x)^2} \, \mathrm{d}t > 0.$$

Hence, $\mathcal{H}[\rho]$ is strictly increasing on $(-\infty, \lambda_-) \cup (\lambda_+, \infty)$. A direct sign check of the integrand confirms that $\mathcal{H}[\rho](x) > 0$ for $x < \lambda_-$ and $\mathcal{H}[\rho](x) < 0$ for $x > \lambda_+$, establishing items (1) and (3). Furthermore, the asymptotic statement in item (4) follows directly from (King, 2009, Eq. 4.111), which gives $\mathcal{H}[\rho](x) \sim -(\int \rho \, \mathrm{d}x)/x$ as $|x| \to \infty$; since $\int \rho \, \mathrm{d}x = 1$, this is $-1/x$.

*Step 2. Existence of a zero.* From steps (1), we have $\mathcal{H}[\rho] > 0$ as $x \downarrow \lambda_-$, and $\mathcal{H}[\rho] < 0$ as $x \uparrow \lambda_+$. Since $\mathcal{H}[\rho]$ is continuous—guaranteed by by the boundedness of $\rho$—the intermediate value

theorem ensures that there exists at least one root $x^* \in (\lambda_-, \lambda_+)$ such that $\mathcal{H}[\rho](x^*) = 0$. Moreover, every simple zero flips the sign of $\mathcal{H}[\rho]$, so starting with positive sign on the left and finishing with negative sign on the right forces an odd number of sign-changes and hence an odd number of interior zeros. $\qquad \square$

The properties in Proposition C.3 provide a clear picture of the behavior of the Hilbert transform of a probability density with compact support; see Figure C.1 for an example.

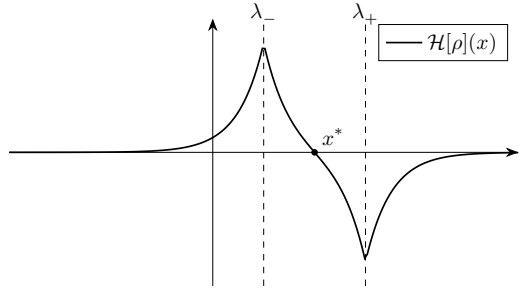

Figure C.1: A schematic example of the behavior of the Hilbert transform as prescribed by Proposition C.3.

We now seek a practical approximation for the real part of the glue function $G(x)$ on the support interval $I$. Motivated by the fact that many classical spectral laws—such as the free Meixner family of distributions in Appendix B.1—admit closed-form Hilbert transforms of rational form, we adopt a Padé-type ansatz:

$$G(x) \approx \frac{P(x)}{Q(x)}, \qquad (\text{C.6})$$

with $P$ and $Q$ polynomials of minimal degree. In the classical examples of Appendix B.1, this rational structure is not merely an approximation but *exact*, as shown in (B.2), with $P$ at most linear and $Q$ at most quadratic (see Table B.3).

Even in more general settings—where the density $\rho$ is empirical or lacks a closed-form expression—the Padé approximant remains attractive as it provides a compact, flexible representation with interpretable parameters. Once we commit to this functional form, we can leverage structural properties of the Hilbert transform to constrain the numerator and denominator more effectively. Proposition C.3 provides key analytic features of $\mathcal{H}[\rho]$ that guide this design. Namely, $P$ should have at least a zero inside $I$, while $Q$ should have no zeros in $\mathbb{C}^-$ or within $I$, though it may have zeros in $\mathbb{R} \setminus I$. Mirroring these features, we parametrize the Padé approximant as

$$G(z) = \frac{P(z)}{Q(z)} = d + cz + \sum_{j=1}^{q} \frac{r_i}{z - a_j},$$

which is a Padé approximant of degree $[q + 1/q]$, or $[q/q]$ if $c = 0$ or $[q - 1/q]$ if $c = d = 0$. We constrain the parameters to ensure there is at least one zero in $I$ and no pole $a_j$ inside $I$.

## Appendix D    Methods of Estimating Stieltjes Transform

Given a probability density $\rho(x)$ supported on an interval $I \subset \mathbb{R}$, its Stieltjes transform is defined by

$$m(z) := \int_{\mathbb{R}} \frac{\rho(x)}{x - z} \, \mathrm{d}x, \qquad z \in \mathbb{C} \setminus I.$$

The density $\rho$ can be recovered from the boundary behavior of $m(z)$ via the inversion formula:

$$\rho(x) = \frac{1}{\pi} \lim_{\epsilon \to 0^+} \Im \, m(x + i\epsilon).$$

Accurately estimating $m(z)$ is therefore essential for recovering the underlying spectral density. Given a Hermitian matrix $\mathbf{A} \in \mathbb{R}^{n \times n}$, the Stieltjes transform of its empirical spectral distribution is

defined by

$$m(z) = \frac{1}{n} \mathrm{tr}\, (\mathbf{A} - z\mathbf{I})^{-1}, \quad z \in \mathbb{C} \setminus I.$$

A direct way to compute $m(z)$ is

$$m(z) = \frac{1}{n} \sum_{i=1}^{n} \frac{1}{\lambda_i - z},$$

where $\lambda_i$ are the eigenvalues of $\mathbf{A}$. However, because the characteristic curves must pass through the roots of $m$ (Proposition C.2), applying free decompression to this empirical version of $m$ will not remove its poles. It is necessary to consider approximations of $m$ that correspond to Stieltjes transforms for smooth densities.

In this section, we describe three numerical strategies for estimating the Stieltjes transform of a Hermitian matrix. The most widely used approach in the literature is based on the Lanczos algorithm (Appendix D.1), which constructs a continued fraction approximation to the resolvent. However, we find that this method often suffers from slow convergence and numerical instability, due in part to its reliance on a random initial vector and the large number of Lanczos steps required to achieve acceptable accuracy. As such, it is often ill-suited for tasks that demand high-precision estimates of $m(z)$.

To overcome these limitations, we present two alternative methods. The first, described in Appendix D.2, expands the spectral density in a Jacobi polynomial basis, offering flexibility in modeling various edge behaviors. The second, described in Appendix D.3, specializes this approach to Chebyshev polynomials, for which we derive a closed-form expression for the Stieltjes transform of each basis element. This enables direct and stable evaluation via power series. Together, these methods offer more accurate and robust alternatives to Lanczos-based approaches, especially in regimes where high-resolution spectral information is required.

### D.1 Estimating Stieltjes Transform using Lanczos Method

The key idea of the Lanczos method is to approximate $m(z)$ by constructing a rational function—specifically, a continued fraction—that reflects the spectral properties of $\mathbf{A}$. This is achieved via a three-term recurrence relation satisfied by orthogonal polynomials associated with the spectral measure (Golub & Meurant, 2010; Meurant, 2006). The recurrence is obtained from the Lanczos algorithm, which builds a tridiagonal (Jacobi) matrix capturing the action of $\mathbf{A}$ on a low-dimensional Krylov subspace. In brief, the procedure is as follows.

Given a normalized starting vector $\boldsymbol{v}_0 \in \mathbb{R}^n$, $\|\boldsymbol{v}\|_2 = 1$, the Lanczos algorithm constructs an orthonormal basis $\{\boldsymbol{v}_0, \boldsymbol{v}_1, \ldots, \boldsymbol{v}_{p-1}\}$ for the Krylov subspace

$$\mathcal{K}_p(\mathbf{A}, \boldsymbol{v}_0) = \mathrm{span}\left\{\boldsymbol{v}_0, \mathbf{A}\boldsymbol{v}_0, \mathbf{A}^2\boldsymbol{v}_0, \ldots, \mathbf{A}^{p-1}\boldsymbol{v}_0\right\},$$

and produces the symmetric triadiagonal Jacobi matrix

$$\mathbf{T}_p := \begin{bmatrix} \alpha_0 & \beta_1 & & \\ \beta_1 & \alpha_1 & \beta_2 & \\ & \ddots & \ddots & \ddots \\ & & \beta_{p-1} & \alpha_{p-1} \end{bmatrix},$$

which satisfies

$$\mathbf{A}\mathbf{V}_p \approx \mathbf{V}_p\mathbf{T}_p,$$

where $\mathbf{V}_p := [\boldsymbol{v}_0, \boldsymbol{v}_1, \ldots, \boldsymbol{v}_{p-1}] \in \mathbb{R}^{n \times p}$. This identity captures the three-term recurrence relation satisfied by the orthogonal polynomials associated with the spectral measure of $\mathbf{A}$. The Stieltjes transform $m(z)$ can then be approximated by the first element of the resolvent of $\mathbf{T}_p$:

$$m_p(z) = \boldsymbol{e}_1^\mathsf{T}(\mathbf{T}_p - z\mathbf{I})^{-1}\boldsymbol{e}_1.$$

where $\boldsymbol{e}_1 = (1, 0, \ldots, 0) \in \mathbb{R}^n$ denotes the first standard basis vector. The above expression is equivalent to the continued fraction

$$m_p(z) = \cfrac{1}{z - \alpha_0 - \cfrac{\beta_0^2}{z - \alpha_1 - \cfrac{\beta_1^2}{\ddots - \cfrac{\beta_{p-1}^2}{z - \alpha_p}}}}.$$

This approximation is the $[0/p]$ Padé approximant of $m(z)$, and typically exhibits convergence with increasing $p$, especially when $m(z)$ is analytic away from the real axis (Golub & Meurant, 1997). However, the effectiveness of this approximation in practice depends on several implementation details, which we briefly summarize below.

The choice of the starting vector $\boldsymbol{v}_0$ is not critical, as any unit-norm vector suffices. In practice, a randomly chosen unit vector often leads to a convergence, while in theory, the approximation $m_p(z)$ is independent of this choice when $p = n$. However, for larger values of $p$, numerical stability may require full or partial reorthogonalization during the Lanczos process to maintain orthogonality of the Krylov basis vectors (Meurant, 2006).

A typical stopping criterion is based on monitoring the difference between successive approximants and terminating when

$$\left| m_p(z) - m_{p-1}(z) \right| < \varepsilon,$$

for a fixed test point $z \in \mathbb{C}^+$; see (Frommer & Schweitzer, 2016; Ubaru et al., 2017) for detailed analyses of convergence rates and practical recommendations.

The continued fraction $m_p(z)$ can be evaluated efficiently either through recursive computation or, equivalently, by solving the tridiagonal system $(\mathbf{T}_p - z\mathbf{I})\boldsymbol{x} = \boldsymbol{e}_1$. The overall computational cost is $\mathcal{O}(pn^2)$ for dense matrices or $\mathcal{O}(p \operatorname{nnz}(\mathbf{A}))$ for sparse matrices (where $\operatorname{nnz}(\mathbf{A})$ is the number of non-zero elements of $\mathbf{A}$), plus $\mathcal{O}(p^2)$ for reorthogonalization. If the algorithm is run to completion with $p = n$ and numerical orthogonality is preserved, the approximation yields the exact spectrum of $\mathbf{A}$, and thus fully recovers $m(z)$.

### D.2 Estimating Stieltjes Transform using Jacobi Polynomials

Jacobi polynomials provide a highly flexible basis for approximating spectral densities, particularly when the distribution exhibits distinctive edge behavior—such as square-root singularities or vanishing derivatives at the boundaries of the support. This family of orthogonal polynomials can represent a wide range of distributions encountered in random matrix theory (e.g., distributions in Appendix B.1) and more general beta-like distributions.

Let $\rho(x)$ be a probability density with support on the interval $[\lambda_-, \lambda_+]$. Through an affine change of variables, we map the domain to the canonical interval $[-1, 1]$ via

$$t(x) := \frac{2x - (\lambda_+ + \lambda_-)}{\lambda_+ - \lambda_-}.$$

The density is then expanded in the Jacobi polynomial basis:

$$\rho(x) \approx \sum_{k=0}^{K} \psi_k \, w^{(\alpha,\beta)}(t(x)) \, P_k^{(\alpha,\beta)}(t(x)), \tag{D.1}$$

where $P_k^{(\alpha,\beta)}$ is the Jacobi polynomial of degree $k$ and parameters $\alpha, \beta > -1$, orthogonal with respect to the weight

$$w^{(\alpha,\beta)}(t) = (1 - t)^\alpha (1 + t)^\beta, \qquad \alpha, \beta > -1.$$

This expansion is especially effective when $\rho(x) \sim (\lambda_+ - x)^\alpha (x - \lambda_-)^\beta$ near the endpoints, aligning the basis with the singularity structure of $\rho$, and accelerating the convergence.

**Projection.** Given a set of eigenvalues $\{\lambda_i\}_{i=1}^N$, we first construct a smoothed density estimate $\hat{\rho}(x)$, e.g., via Gaussian or beta kernel density estimation. Mapping to the variable $t$, we approximate the expansion coefficients using the orthogonality of Jacobi polynomials:

$$\psi_k = \frac{1}{\|P_k^{(\alpha,\beta)}\|^2} \int_{-1}^1 \hat{\rho}(t) P_k^{(\alpha,\beta)}(t)\,dt \approx \frac{\sum_j \hat{\rho}(t_j) P_k^{(\alpha,\beta)}(t_j)\Delta t}{\|P_k^{(\alpha,\beta)}\|^2}, \qquad (\text{D.2})$$

where $\{t_j\}$ are grid points on $[-1,1]$ and $\|P_k^{(\alpha,\beta)}\|^2$ denotes the squared norm, which admits a closed-form expression

$$\|P_k^{(\alpha,\beta)}\|^2 = \frac{2^{\alpha+\beta+1}}{2k+\alpha+\beta+1} \frac{\Gamma(k+\alpha+1)\,\Gamma(k+\beta+1)}{\Gamma(k+1)\,\Gamma(k+\alpha+\beta+1)}.$$

In the above, $\Gamma$ is the Gamma function.

**Stieltjes transform in Jacobi basis.** Once the coefficients $\psi_k$ are obtained, the Stieltjes transform

$$m(z) = \int_{\lambda_-}^{\lambda_+} \frac{\rho(x)}{x-z}\,dx,$$

can be approximated via the expansion (D.1). First, by mapping $x$ and $z$ as

$$x(t) := \frac{1}{2}(\lambda_+ + \lambda_-) + \frac{1}{2}(\lambda_+ - \lambda_-)t, \qquad t \in [-1,1],$$

$$z(u) := \frac{1}{2}(\lambda_+ + \lambda_-) + \frac{1}{2}(\lambda_+ - \lambda_-)u, \qquad u \in \mathbb{C} \setminus [-1,1],$$

we arrive at the transformed expression

$$m(u) \approx \frac{2}{\lambda_+ - \lambda_-} \sum_{k=0}^K \psi_k \int_{-1}^1 \frac{w^{(\alpha,\beta)}(t) P_k^{(\alpha,\beta)}(t)}{t-u}\,dt, \quad u \in \mathbb{C}^+ \setminus [-1,1].$$

The inner integral in the above defines the so-called *Jacobi function of the second kind*,

$$Q_k^{(\alpha,\beta)}(u) = \int_{-1}^1 \frac{w^{(\alpha,\beta)}(t) P_k^{(\alpha,\beta)}(t)}{t-u}\,dt,$$

and the Stieltjes transform becomes

$$m(z(u)) \approx \frac{2}{\lambda_+ - \lambda_-} \sum_{k=0}^K \psi_k\, Q_k^{(\alpha,\beta)}(u).$$

**Gauss–Jacobi quadrature.** In practice, we do not evaluate $Q_k^{(\alpha,\beta)}$ directly via numerical integration. Instead, we compute each term using a dedicated Gauss–Jacobi quadrature rule (Shen et al., 2011, Section 3.2.2) tailored to the weight $w^{(\alpha,\beta)}$

$$Q_k^{(\alpha,\beta)}(u) \approx \sum_{i=1}^{n_k} \frac{w_i P_k^{(\alpha,\beta)}(t_i)}{t_i - u},$$

where $\{t_i, w_i\}$ are the quadrature nodes and weights for the given weight function. The number of nodes $n_k$ depends on the degree $k$, typically chosen as $n_k := \max(k+1, n_0)$. Evaluating each mode $Q_k^{(\alpha,\beta)}(u)$ independently using its own quadrature is critical for numerical stability, especially for large $k$. This separation of summation and integration avoids artifacts that arise when attempting to quadrature-integrate the full series at once.

**Choosing parameters.** Jacobi polynomials provide a flexible basis for modeling spectral densities with various edge behaviors, particularly those of the form $\rho(x) \sim (\lambda_+ - x)^\alpha (x - \lambda_-)^\beta$. Different choices of $\alpha$ and $\beta$ yield classical orthogonal polynomials:

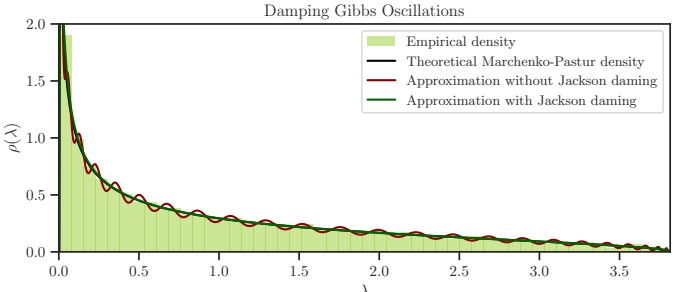

Figure D.1: Effect of Jackson damping on approximating the Marchenko–Pastur density from the Gram matrix $\frac{1}{d}\mathbf{X}\mathbf{X}^\intercal$, where $\mathbf{X}$ is an $n \times d$ random matrix with $n = 1000$ and $n/d = 0.9$. Parameters were intentionally selected to amplify Gibbs oscillations for illustrative purposes; typical cases exhibit much milder artifacts.

- $\alpha = \beta = 0$ recovers Legendre polynomials.

- $\alpha = \beta = \frac{1}{2}$ gives Chebyshev polynomials of the second kind, well-suited for densities with square-root vanishing edges, such as $\rho(x) \sim \sqrt{(\lambda_+ - x)(x - \lambda_-)}$, which are common in free probability models (see Table B.1).

- $\alpha = \beta = -\frac{1}{2}$ yields Chebyshev polynomials of the first kind, appropriate for densities with inverse square-root singularities at the edges.

This parameterization aligns the orthogonal basis with the singular structure of $\rho(x)$, often resulting in exponential convergence with respect to the number of terms $K$. In practice, we find that the method is not highly sensitive to precise values of $\alpha$ and $\beta$, as long as their qualitative effect on edge behavior is appropriate.

**Gibbs oscillations.** Expansions in orthogonal polynomials can exhibit Gibbs oscillations, especially for large truncation orders $K$. These oscillations can be suppressed by applying damping to the coefficients via a filter $g_k$, resulting in the smoothed expansion:

$$\rho(x) \approx \sum_{k=0}^{K} g_k \psi_k P_k^{(\alpha,\beta)}(x).$$

Several damping schemes exist, including Jackson (1912), Lanczos, Fejér, and Parzen filters. In this work, we use Jackson damping, defined as

$$g_k := \frac{(K - k + 1)\cos\left(\frac{\pi k}{K+1}\right) + \sin\left(\frac{\pi k}{K+1}\right)\cot\left(\frac{\pi}{K+1}\right)}{K + 1}, \quad k = 0, 1, \ldots, K.$$

Figure D.1 shows how Jackson damping improves the stability of the approximation.

**Regularization.** To further stabilize the coefficients $\psi_k$, we introduce Tikhonov regularization by modifying (D.2) as

$$\psi_k = \frac{1}{\gamma_k + \|P_k^{(\alpha,\beta)}\|^2} \int_{-1}^{1} \hat{\rho}(t) P_k^{(\alpha,\beta)}(t)\, dt,$$

with the penalty term $\gamma_k$ chosen as

$$\gamma_k := \gamma \left(\frac{k}{K+1}\right)^2,$$

where $\gamma > 0$ is a small regularization parameter. This form penalizes higher-order coefficients more strongly, resulting in smoother approximations without significantly biasing the shape or integral of the estimated density.

**Preserving positivity and mass.**  Finally, to enforce positivity of $\rho(x)$ over its support and ensure $\int \rho(x)\,\mathrm{d}x = 1$, we perform a secondary constrained optimization over $\psi_k$, initialized at the projected values, subject to constraints enforcing unit mass and pointwise non-negativity over a fine grid.

## D.3   Estimating Stieltjes Transform using Chebyshev Polynomials

Constructing the principal branch of the Stieltjes transform numerically is straightforward: one can estimate $m$ on the real axis by letting $m(x + i0^+) = \mathcal{H}[\rho](x) + i\pi\rho(x)$ and estimating the Hilbert transform using FFT techniques. Then, $m$ can be extended to $\mathbb{C}^+$ by convolution with the Poisson kernel. The lower-half plane is obtained by the Schwarz reflection principle. However, constructing the secondary branch of the Stieltjes transform numerically is ordinarily very challenging, as it is tantamount to numerical analytic continuation.

Our primary mechanism for estimating the Stieltjes transform of a density with compact support is the following lemma. Here, we let $j(z) := \frac{1}{2}(z + z^{-1})$ denote the Joukowski transform, and let

$$J(z) := z - \sqrt{z^2 - 1}$$

denote the inverse Joukowski transform, where we take the branch of the square root to have positive imaginary part for $z \in \mathbb{C}^+$. The Chebyshev polynomials of the second kind $U_k(x)$ are defined for $x \in [-1, 1]$ by the recurrence relation

$$U_{k+1}(x) = 2xU_k(x) - U_{k-1}(x)$$

from $U_0(x) = 1$ and $U_1(x) = 2x$.

**Lemma D.1.** *The Stieltjes transform of $U_k(x)\sqrt{1 - x^2}$ on $[-1, 1]$ is given by $\pi J(z)^{k+1}$ for any $k \in \{0\} \cup \mathbb{N}$.*

**Proof.** Let $m(z) = \pi J(z)^{k+1}$. Since $m : \mathbb{C}^+ \to \mathbb{C}^+$ is analytic, by the Herglotz-Nevanlinna representation theorem, there exist constants $C, D > 0$ and a Borel measure $\mu$ such that

$$m(z) = C + Dz + \int \frac{1}{\lambda - z}\mathrm{d}\mu(\lambda).$$

Since $J(i\eta) \to 0$ as $\eta \to \infty$, $C = D = 0$, and so $m$ is the Stieltjes transform of a measure $\mu$. If $x := \cos\theta$ for $\theta \in [0, \pi]$, since $j(e^{i\theta}) = \cos\theta$, $j(e^{i\theta}) = x$ and $J(x) = e^{i\theta}$. Since $U_k(\cos\theta)\sin\theta = \sin((k + 1)\theta)$, it follows that

$$U_k(x)\sqrt{1 - x^2} = \Im(e^{i(k+1)\theta}) = \Im(J(x)^{k+1}).$$

Since $J$ is analytic on $\mathbb{C}^+$, this, in turn, implies that

$$\frac{\mathrm{d}\mu(x)}{\mathrm{d}x} = \lim_{\delta \to 0^+} \frac{1}{\pi}\Im[m(x + i\delta)] = U_k(x)\sqrt{1 - x^2},$$

and the result follows. $\qquad\square$

The significance of Lemma D.1 is that we can approximate a spectral density as a Chebyshev series, and immediately read off the corresponding Stieltjes transform. To see this, suppose that a spectral density $\rho$ on $[-1, 1]$ has the Chebyshev expansion

$$\rho(x) = \sqrt{1 - x^2}\sum_{k=0}^{\infty}\psi_k U_k(x).$$

By Fubini's Theorem,

$$\mathbb{E}_{X \sim \rho}[U_k(X)] = \sum_{l=0}^{\infty}\psi_l\int_{-1}^{1}U_k(x)U_l(x)\sqrt{1 - x^2}\,\mathrm{d}x = \frac{\pi}{2}\psi_k,$$

and so $\psi_k$ can be estimated by $\hat{\psi}_{n,k} = \frac{2}{\pi n} \sum_{i=1}^{n} U_k(\lambda_i)$. The corresponding Stieltjes transform is then $m(z) = \pi \Lambda(J(z))$ where

$$\Lambda(z) = \sum_{k=0}^{\infty} \psi_k z^{k+1}.$$

Noting that $|J(z)| = \exp(-\Re(\mathrm{arccosh}(z)))$, for $z \in \mathbb{C}^+$, $|J(z)| \leq 1$, and so $J(z)$ will lie in the domain of convergence of the power series for $\Lambda$. However, this series is unlikely to converge for $z \in \mathbb{C}^-$, and is therefore insufficient for estimating the secondary branch. The solution is to analytically continue $\Lambda$ outside the domain of convergence of its power series by appealing to Padé approximants. Computing Padé approximants algebraically is an expensive task, so we appeal to Wynn's $\epsilon$-algorithm to rapidly evaluate the Padé approximant for $\Lambda$, given estimates of its coefficients $\psi_k$ (Brezinski, 1996). For a sequence of partial sums $\Lambda_n(z) = \sum_{k=0}^{n} \psi_k z^{k+1}$, $n = 0, 1, \ldots$, the $\epsilon$-algorithm constructs a tableau $\{\epsilon_k^{(n)}(z)\}_{n,k}$ by

$$\epsilon_{-1}^{(n)}(z) = 0, \quad \epsilon_0^{(n)}(z) = \Lambda_n(z), \quad \epsilon_{k+1}^{(n)}(z) = \epsilon_{k-1}^{(n+1)}(z) + (\epsilon_k^{(n+1)}(z) - \epsilon_k^{(n)}(z))^{-1}.$$

The coefficient $\epsilon_{2k}^{(n)}(z)$ provides the Padé approximant for $\Lambda$ with numerator and denominator degrees of $k + n$ and $k$, respectively (Baker Jr. & Graves-Morris, 1996, Eq. (4.10)). In particular, Nuttal's Theorem (Baker Jr. & Graves-Morris, 1996, Theorem 6.5.3) shows that $\epsilon_{2n}^{(0)}(z)$ converges in measure to the analytic continuation of $\Lambda(z)$ *regardless* of whether the partial sums $\Lambda_n$ converge.

---

**Algorithm 2:** Padé–Chebyshev Free Decompression

---

**Input** : Hermitian matrix $\mathbf{A} \in \mathbb{R}^{n \times n}$;
Maximum Chebyshev polynomial order $K$;
Evaluation set $\mathcal{X} \subset \mathbb{R}$;
Subsampling parameter $n_s$ (with $1 \leq n_s \leq n$);
Perturbation $\delta$ where $0 < \delta \ll 1$.

**Output** : Estimated spectral density values $\{\hat{\rho}(x)\}_{x \in \mathcal{X}}$.

*// Step 1: Generate eigenvalues from a random submatrix*

**1** Extract a random submatrix $\hat{\mathbf{A}}$ from $\mathbf{A}$ and compute its eigenvalues $\{\lambda_i\}_{i=1}^{n}$.

**2** $\lambda_- \leftarrow \min\{\lambda_i\}_{i=1}^{n} - \delta n^{-1}$                 *// Estimate of the left endpoint of the support*

**3** $\lambda_+ \leftarrow \max\{\lambda_i\}_{i=1}^{n} + \delta n^{-1}$                 *// Estimate of the right endpoint of the support*

*// Step 2: Estimate Chebyshev coefficients*

**4 for** $k = 0$ **to** $K$ **do**

**5**      $\hat{\psi}_{n,k} \leftarrow \frac{4}{\pi(\lambda_+ - \lambda_-)n} \sum_{i=1}^{n} U_k(M_{\lambda_-,\lambda_+}(\lambda_i))$

*// Step 3: Compute the Padé approximant*

**6** Using Wynn's $\epsilon$–algorithm, obtain the Padé approximant $\Lambda(z)$ for the power series $\sum_{k=0}^{K} \hat{\psi}_{n,k} z^{k+1}$.

*// Step 4: Form the Stieltjes transform*

**7** $m(z) \leftarrow \pi \Lambda(J(M_{a,b}(z)))$

*// Step 5: Solve characteristic curves*

**8 foreach** $x \in \mathcal{X}$ **do**

**9**      Using Newton's method, find $z_x \in \mathbb{C}$ such that

$$x + i\delta = z_x - \frac{\frac{n}{n_s} - 1}{m(z_x)}.$$

*// Step 6: Return the estimated spectral density*

**10 return** $\left\{ \frac{1}{\pi} \Im\, m(z_x) \frac{n_s}{n} \right\}_{x \in \mathcal{X}}$.

---

To extend this procedure to an arbitrary support interval $[\lambda_-, \lambda_+]$, we can make use of a conformal map

$$M_{\lambda_-, \lambda_+}(z) = \frac{2z - (\lambda_+ + \lambda_-)}{\lambda_+ - \lambda_-} \quad : \quad [\lambda_-, \lambda_+] \mapsto [-1, 1].$$

In this case, the spectral density is given by

$$\rho(x) = \frac{2\sqrt{(\lambda_+ - x)(x - \lambda_-)}}{b - a} \sum_{k=0}^{\infty} \psi_k U_k(M_{\lambda_-, \lambda_+}(x)),$$

where the coefficients of the estimates are

$$\hat{\psi}_{n,k} = \frac{4}{\pi n(\lambda_+ - \lambda_-)} U_k(\lambda_i),$$

and the Stieltjes transform becomes

$$m(z) = \pi \Lambda(J(M_{\lambda_-, \lambda_+}(z))).$$

## Appendix E   Proofs of Propositions 2 and 3

**Proof of Proposition 2.** By construction, $R(t, e^{-t}z) = R(0, z)$. In terms of the inverse Stieltjes transform, this implies

$$\omega(t, e^{-t}z) - \left(1 - e^t\right)\frac{1}{z} = \omega(0, z).$$

Differentiating both sides in $t$ gives

$$\frac{\partial \omega}{\partial t}(t, e^{-t}z) - ze^{-t}\frac{\partial \omega}{\partial z}(t, e^{-t}z) + \frac{e^t}{z} = 0. \tag{E.1}$$

Since $m(t, \omega(t, z)) = z$, $\frac{\partial m}{\partial z}\frac{\partial \omega}{\partial z} = 1$ and $\frac{\partial m}{\partial t} + \frac{\partial m}{\partial z}\frac{\partial \omega}{\partial t} = 0$. Applying these relations to (E.1) gives

$$-\frac{\partial m}{\partial t}(t, \omega(t, e^{-t}z)) - ze^{-t} + \frac{e^t}{z}\frac{\partial m}{\partial z}(t, \omega(t, e^{-t}z)) = 0.$$

Now evaluate this expression at $z = \tilde{z}$ where $\tilde{z} = e^t m(t, z)$ so that $\omega(t, e^{-t}\tilde{z}) = z$. Then

$$-\frac{\partial m}{\partial t}(t, z) - m(t, z) + \frac{1}{m(t, z)}\frac{\partial m}{\partial z}(t, z) = 0.$$

Rearranging this expression implies the result. $\qquad \square$

**Proof of Proposition 3.** Let $\tau \mapsto \big(t(\tau), z(\tau)\big)$ be a characteristic curve of the quasilinear PDE (1). By the chain rule,

$$\frac{\mathrm{d}m(t(\tau), z(\tau))}{\mathrm{d}\tau} = \frac{\partial m(t, z)}{\partial t}\frac{\mathrm{d}t(\tau)}{\mathrm{d}\tau} + \frac{\partial m(t, z)}{\partial z}\frac{\mathrm{d}z(\tau)}{\mathrm{d}\tau}. \tag{E.2}$$

Matching coefficients in (E.2) with the PDE (1) yields the system of ODEs

$$\frac{\mathrm{d}t(\tau)}{\mathrm{d}\tau} = 1, \qquad\qquad t(0) = 0, \tag{E.3a}$$

$$\frac{\mathrm{d}z(\tau)}{\mathrm{d}\tau} = -m(\tau)^{-1}, \qquad z(0) = z_0, \tag{E.3b}$$

$$\frac{\mathrm{d}m(\tau)}{\mathrm{d}\tau} = -m(\tau), \qquad m(0) = m_0(z_0). \tag{E.3c}$$

Solving (E.3) is straightforward, yielding (2).

By standard ODE existence and uniqueness theorems, for each $z_0$ there is a unique characteristic curve $\tau \mapsto (t(\tau), z(\tau), m(\tau))$ near $\tau = 0$. Since $\frac{dt}{d\tau} = 1$ and $\frac{dz}{d\tau} \neq 0$ if $m_0(z_0) \neq 0$, we can (locally) invert $\tau \mapsto t$ and $(z_0) \mapsto z$. This yields a unique $C^1$ function $m(t, z)$ solving (1) with $m(0, z) = m_0(z)$.

Finally, the condition $m_0(z) \neq 0$ for all $z \in \Omega$ ensures that $m(\tau) \neq 0$ in a neighborhood of $\tau = 0$, so the term $m^{-1}$ in (1) is well-defined. Uniqueness follows from the uniqueness of solutions to (E.3) with given initial data. We see that Proposition 3 gives explicit solutions to (1) in terms of the initial data and desired time $t$.

It remains to show the existence of $\phi$. Define $\Phi(t, z, z_0) := z_0 + m_0(z_0)^{-1}(1 - e^t) - z$. If $\partial \Phi / \partial z_0 \neq 0$, by the implicit function theorem (Hamilton, 1982), there exists an inverse function $z_0 = \phi(t, z)$ solving $\Phi(t, z, \phi(t, z)) = 0$, which allows the explicit solution of (1) by $m(t, z) = m_0(\phi(t, z))e^{-t}$. This completes the proof. $\qquad\square$

## Appendix F  Stability Analysis

To analyze the stability of $m(t, z)$ over time $t$, we perturb the PDE (1) by replacing $m$ with $m + \delta m$, and derive a PDE governing the evolution of the perturbation field $\delta m(t, z)$. We then quantify the impact of this perturbation using an appropriate functional norm.

A natural choice for this purpose is the $L^2$-norm along horizontal slices in the upper half-plane, defined by

$$\|m(t, \cdot + iy)\|_{L^2(\Gamma_y)}^2 := \int_{\mathbb{R}} |m(t, x + iy)|^2 \, \mathrm{d}x,$$

where $\Gamma_y := \{z = x + iy \mid x \in \mathbb{R}\}$ is the horizontal line at height $y > 0$. This norm is particularly suitable for our analysis since we are ultimately interested in evaluating $m(t, x + i\delta)$ slightly above the real axis, in order to recover the spectral density from the Stieltjes transform.

Moreover, this space includes the boundary values of Stieltjes transforms $m(t, x + i0^+)$ whenever $\rho(t, \cdot) \in L^2(\mathbb{R})$, since one can show that

$$\|m(t, x + i0^+)\|_{L^2(\Gamma_{0^+})}^2 = (1 + \pi^2)\|\rho(t, \cdot)\|_{L^2(\mathbb{R})}^2.$$

We now establish an upper bound on the growth of this norm over time, culminating in a Grönwall-type inequality. This is formalized in the following proposition.

**Proposition F.1.** *Suppose $\rho(t, \cdot) \in L^2(\mathbb{R})$, and let $m(t, z)$ be the Stieltjes transform defined on the domain $\mathbb{C}^+$. Let $\delta m$ be a small functional (Gâteaux) perturbation of $m$ (so $\|\delta m\| \ll 1$) satisfying the Bernstein inequality*

$$\|\partial_x \delta m(t, z)\|_{L^2(\Gamma_y)} \leq M\|\delta m(t, z)\|_{L^2(\Gamma_y)}, \quad z \in \Gamma_y, \forall y > 0 \tag{F.1}$$

*which means on each horizontal slice, $\delta m(t, \cdot + iy)$ is $M$-band-limited (Paley–Wiener space), ensuring sufficient decay of the perturbations. Then, for every $y > 0$, it holds that*

$$\frac{\mathrm{d}}{\mathrm{d}t}\|\delta m(t)\|_{L^2(\Gamma_y)}^2 \leq (-2 + C_y(t)) \|\delta m(t)\|_{L^2(\Gamma_y)}^2, \tag{F.2}$$

*where*

$$C_y(t) := 3 \sup_{z \in \Gamma_y} \left| \Re \frac{\partial_z m(t, z)}{m(t, z)^2} \right| + 2M \sup_{z \in \Gamma_y} \left| \Im \frac{1}{m(t, z)} \right|. \tag{F.3}$$

**Proof.** Let $\tilde{m} = m + \delta m$, where $\delta m \in H^2(\mathbb{R}_{>0} \times \mathbb{C}^+)$ (the usual Hardy space) is a small perturbation with the boundary condition $\delta m(t, z) = 0$ at $|z| \to \infty$. Plugging $\tilde{m}$ into the PDE (1) and subtracting the unperturbed PDE gives the perturbation equation

$$\frac{\partial \delta m}{\partial t} = -\delta m + \frac{1}{m + \delta m} \frac{\partial(m + \delta m)}{\partial z} - \frac{1}{m} \frac{\partial m}{\partial z}.$$

Expanding the nonlinear term using a first-order approximation and rearranging, we obtain

$$\frac{\partial \delta m}{\partial t} = \frac{1}{m} \frac{\partial \delta m}{\partial z} - \delta m - \frac{\delta m}{m^2} \frac{\partial m}{\partial z} + \mathcal{O}(|\delta m|^2).$$

The linearized equation above is essentially a transport-type PDE, with the first, second, and third terms on the right-hand side representing the transport term, damping term, and perturbation term, respectively.

To obtain the energy, we multiply both sides of the above by $\overline{\delta m}$ (the complex conjugate of $\delta m$), integrate over $\Gamma_y$, and consider its real part, yielding

$$\Re \int_{\mathbb{R}} \frac{\partial \delta m}{\partial t} \overline{\delta m} \, \mathrm{d}x = \Re \int_{\mathbb{R}} \frac{1}{m} \frac{\partial \delta m}{\partial z} \overline{\delta m} \, \mathrm{d}x - \Re \int_{\mathbb{R}} \left( 1 + \frac{1}{m^2} \frac{\partial m}{\partial z} \right) |\delta m|^2 \, \mathrm{d}x. \tag{F.4}$$

The term on the left-hand side of (F.4), after taking $\sup_{y>0}$, can be expressed as

$$\Re \int_{\mathbb{R}} \frac{\partial \delta m}{\partial t} \overline{\delta m} \, \mathrm{d}x = \frac{1}{2} \frac{\mathrm{d}}{\mathrm{d}t} \|\delta m\|_{L^2(\Gamma_y)}^2. \tag{F.5}$$

Since $\delta m$ is holomorphic, we have $\partial_z \delta m = \partial_x \delta m$ on every stripe $\Gamma_y$. Writing $m^{-1} = a + ib$ with real valued $a, b$, we have

$$\Re \left( \frac{1}{m} (\partial_x \delta m) \overline{\delta m} \right) = \frac{1}{2} \Re \left( \frac{\partial_x m}{m^2} \right) |\delta m|^2 - b \Im \left( (\partial_x \delta m) \overline{\delta m} \right) + \frac{1}{2} \partial_x (a |\delta m|^2).$$

Integrating over $x \in \mathbb{R}$, the third term on the right hand side of the above vanishes. As for the other terms, we apply Cauchy–Schwarz and the Bernstein bound (F.1) to obtain

$$\Re \int_{\mathbb{R}} \frac{1}{m} \frac{\partial \delta m}{\partial z} \overline{\delta m} \, \mathrm{d}x \leq \left( \frac{1}{2} \sup_{z \in \Gamma_y} \left| \Re \frac{\partial_z m}{m^2} \right| + M \sup_{z \in \Gamma_y} \left| \Im \frac{1}{m} \right| \right) \int_{\mathbb{R}} |\delta m|^2 \, \mathrm{d}x. \tag{F.6}$$

Also, the second term on the right-had side of (F.4) satisfies

$$-\Re \int_{\mathbb{R}} \left( 1 + \frac{1}{m^2} \frac{\partial m}{\partial z} \right) |\delta m|^2 \, \mathrm{d}x \leq -\int_{\mathbb{R}} |\delta m|^2 \, \mathrm{d}x + \sup_{z \in \Gamma_y} \left| \frac{\partial_z m}{m^2} \right| \int_{\mathbb{R}} |\delta m|^2 \, \mathrm{d}x. \tag{F.7}$$

Putting all together by substituting (F.5), (F.6), and (F.7) into (F.4), we obtain

$$\frac{1}{2} \frac{\mathrm{d}}{\mathrm{d}t} \|\delta m\|_{L^2(\Gamma_y)}^2 \leq \left( -1 + \frac{3}{2} \sup_{z \in \Gamma_y} \left| \Re \frac{\partial_z m}{m^2} \right| + M \sup_{z \in \Gamma_y} \left| \Im \frac{1}{m} \right| \right) \int_{\mathbb{R}} |\delta m|^2 \, \mathrm{d}x$$
$$= (-1 + C_y(t)) \|\delta m\|_{L^2(\Gamma_y)}^2.$$

This completes the proof. $\qquad \square$

**Remark F.1.** To complete Proposition F.1, we verify that the constant $C_y(t)$ is finite for every $0 < y < \infty$. Recall the representation of the Stieltjes transform,

$$m(t, z) = \int_{\mathbb{R}} \frac{\rho(t, \xi)}{\xi - z} \, \mathrm{d}\xi, \tag{F.8}$$

for $z = x + iy \in \mathbb{C}^+$ with $y > 0$. Then, since $\rho(t, x)$ is supported on a compact interval of diameter $D := \mathrm{diam}(\mathrm{supp}(\rho))$, we have

$$|m(t, z)| \geq \Im m(t, z) = \int_{\mathbb{R}} \frac{y \, \rho(t, \xi)}{(\xi - x)^2 + y^2} \, \mathrm{d}\xi \geq \frac{y}{D^2 + y^2} \int_{\mathbb{R}} \rho(t, \xi) \, \mathrm{d}\xi = \frac{y}{D^2 + y^2}.$$

It follows that

$$\sup_{z \in \Gamma_y} |\Im m(t, z)^{-1}| \leq \sup_{z \in \Gamma_y} |m(t, z)^{-1}| \leq \frac{D^2 + y^2}{y}. \tag{F.9}$$

On the other hand, from the derivative of (F.8):

$$\partial_z m(t, z) = \int_{\mathbb{R}} \frac{\rho(t, \xi)}{(\xi - z)^2} \, \mathrm{d}\xi,$$

so

$$|\partial_z m(t, z)| \leq \int_{\mathbb{R}} \frac{\rho(t, \xi)}{(\xi - x)^2 + y^2} \, \mathrm{d}\xi \leq \frac{1}{y^2}.$$

Combining the above with (F.9), we obtain

$$\sup_{z \in \Gamma_y} \left| \Re \frac{\partial_z m(t,z)}{m(t,z)^2} \right| \le \sup_{z \in \Gamma_y} \left| \frac{\partial_z m(t,z)}{m(t,z)^2} \right| \le \frac{1}{y^2} \left( \frac{D^2 + y^2}{y} \right)^2 = \frac{(D^2 + y^2)^2}{y^4}. \tag{F.10}$$

Substituting the uniform bounds (F.9) and (F.10) into (F.3) confirms that $C_y(t) < \infty$ for all $0 < y < \infty$.

We note that this bound is not defined at $y = 0$, which is expected: as $y \to 0^+$, the Stieltjes transform approaches the real axis where it develops discontinuities, and quantities like $\partial_z m$ may diverge. However, in both our theory and implementation, we evaluate $m(t,z)$ slightly above the real axis (e.g., at $z = x + i\delta$ for fixed $0 < \delta \ll 1$), where all terms remain finite. In practice, the observed error propagation is significantly milder than this worst-case upper bound suggests. △

Proposition F.1 then implies that the perturbation $\delta m$ satisfies an exponential growth bound in time:

$$\|\delta m(t)\|_{L^2(\Gamma_y)}^2 \le \|\delta m_0\|_{L^2(\Gamma_y)}^2 e^{\nu t},$$

with exponent $\nu = -2 + C_y < \infty$ for any fixed $y > 0$. But since $e^t = \frac{n}{n_s}$, the exponential bound translates to polynomial growth in $n$,

$$\|\delta m(t)\|_{L^2(\Gamma_y)}^2 \le \|\delta m_0\|_{L^2(\Gamma_y)}^2 \left( \frac{n}{n_s} \right)^{\nu}.$$

## Appendix G   Background Material for the Neural Tangent Kernel

The neural tangent kernel (NTK) is an important object in machine learning (Jacot et al., 2018), and has become a prominent theoretical and practical tool for studying the behavior of neural networks both during training and inference (Novak et al., 2022). The Gram matrix associated to the NTK has been used in lazy training (Chizat et al., 2019), and as a tool for uncertainty quantification and estimation (Immer et al., 2021; Wilson et al., 2025), exploiting the deep connections between neural networks and Gaussain processes. More recently, it has been shown to arise naturally as a measure of model quality in approximations of the marginal likelihood (Hodgkinson et al., 2023b; Immer et al., 2023), as a tool for quantification of model complexity (Vakili et al., 2022), and in estimation of generalization error in PAC-Bayes bounds (Hodgkinson et al., 2023c; Kim et al., 2023).

While it was first derived in the context of neural networks, the quantity is well-defined for a broader class of functions. For clarity will consider the setting where the underlying function is $C^1$. However, this can be relaxed in practice to include, as an important example, networks with ReLU activations. For such a function $f_{\boldsymbol{\theta}} : \mathcal{X} \to \mathbb{R}^d$ parameterized by $\boldsymbol{\theta} \in \mathbb{R}^p$, we define the (empirical) NTK to be

$$\kappa_{\boldsymbol{\theta}}(x, x') := \mathbf{J}_{\boldsymbol{\theta}}\big(f_{\boldsymbol{\theta}}(x)\big) \mathbf{J}_{\boldsymbol{\theta}}\big(f_{\boldsymbol{\theta}}(x')\big)^{\mathsf{T}}. \tag{G.1}$$

Here, $\mathbf{J}_{\boldsymbol{\theta}}(f_{\boldsymbol{\theta}}(x)) \in \mathbb{R}^{d \times p}$ is the Jacobian of $f_{\boldsymbol{\theta}}$ with respect to its (flattened vector of) parameters $\boldsymbol{\theta}$, evaluated at the data point $x$. Since $\kappa_{\boldsymbol{\theta}}(x, x') \in \mathbb{R}^{d \times d}$, evaluating the NTK over $n$ data points produces a fourth-order tensor of shape $(n, n, d, d)$. For computational convenience, this tensor is typically reshaped into a two-dimensional block matrix of size $nd \times nd$, where each $(i, j)$-block contains the $d \times d$ matrix $\kappa(x_i, x_j)$.

The fact that the NTK arises in a variety of contexts makes it a natural object of study. For well-trained models, the Gram matrix is nearly singular, making accurate characterization of its spectral properties crucial for estimating quantities used in downstream tasks (Ameli et al., 2025). This also motivates considering the spectrum on a logarithmic scale. In Section 5, we analyze the empirical NTK of a ResNet50 model trained on the CIFAR10 dataset, a 10-class classification task. In this case, 10% of the eigenvalues are extremely small ($\sim 10^{-14}$), reflecting near-singular behavior likely tied to the model's class structure.

To examine the scaling behavior of the non-singular bulk spectrum, we project the NTK matrix onto its non-null eigenspace, removing the low-rank null component. We then randomly sample orthogonal eigenvectors within this reduced space and reconstruct a corresponding Hermitian matrix, which is subsequently permuted to generate random subsets for use in our free decompression procedure.

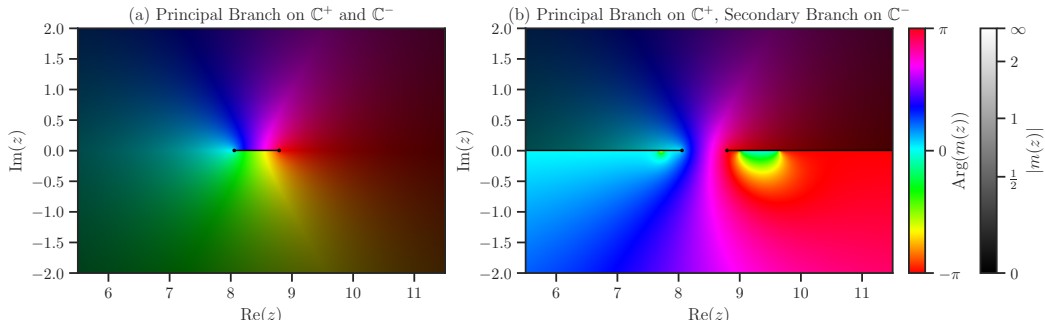

Figure G.1: Analytic continuation of the Stieltjes transform of a density estimate of the spectral distribution of the log-NTK matrix described in Section 5. (a) The principal branch contains a branch cut along the support of the spectral density, while (b) the secondary branch is continuous in this region.

The Stieltjes transform of the approximated spectrum of this matrix is depicted in Figure G.1, which shows its analytic continuation using the principal branch (left), as well as the secondary branch (right).

## Appendix H    Implementation and Reproducibility Guide

We developed a Python package, `freealg`,[3] which implements our algorithms and enables reproduction of all numerical results in this paper. A minimal example using the `freealg.FreeForm` class is shown in Listing 1. This example generates the results in Figures 1 and 2, illustrating the Marchenko–Pastur distribution with parameter $\lambda = \frac{1}{50}$, starting from a matrix of size $n_s = 1000$ and decompressing to a matrix of size $n = 32{,}000$.

Hyperparameters used for the numerical experiments appearing in this document can be found in Table H.1 (notation summarized in the caption). Notebook files that reproduce the figures appearing in this work can be found in the codebase. Further details on function arguments and class parameters are available in the package documentation.

Table H.1: Hyperparameters used in the numerical examples. Notation: $h$ smoothing kernel bandwidth; $(\alpha, \beta)$ Jacobi shape parameters; $K$ Jacobi polynomial order; $\gamma$ regularization strength; Padé $(p, q)$ numerator/denominator degrees; $\delta$ imaginary shift for Stieltjes evaluation (Plemelj–Sokhotski).

| Example | Reference | Smoothing | | Jacobi Polynomials | | | | Stieltjes Transform | |
| --- | --- | --- | --- | --- | --- | --- | --- | --- | --- |
| | | Kernel | $h$ | $(\alpha, \beta)$ | $K$ | $\gamma$ | Damping | Padé $(p, q)$ | $\delta$ |
| Marchenko–Pastur | Figure 1 | Beta | $3 \times 10^{-3}$ | $(\frac{1}{2}, \frac{1}{2})$ | 50 | 0 | Jackson | $(1, 1)$ | $10^{-4}$ |
| Wigner semicircle | Figure B.1 | Beta | $10^{-2}$ | $(\frac{1}{2}, \frac{1}{2})$ | 50 | 0 | Jackson | $(1, 1)$ | $10^{-6}$ |
| Kesten–McKay | Figure B.2 | Beta | $10^{-3}$ | $(\frac{1}{2}, \frac{1}{2})$ | 50 | 0 | Jackson | $(1, 2)$ | $10^{-6}$ |
| Wachter | Figure B.3 | Beta | $10^{-3}$ | $(\frac{1}{2}, \frac{1}{2})$ | 20 | 0 | Jackson | $(1, 2)$ | $10^{-4}$ |
| Meixner | Figure B.4 | Beta | $10^{-3}$ | $(\frac{1}{2}, \frac{1}{2})$ | 50 | 0 | Jackson | $(7, 8)$ | $10^{-6}$ |
| Page-Page | Figure 3 (a) | Beta | $10^{-5}$ | $(0, 0)$ | 100 | 0 | Jackson | $(4, 3)$ | $10^{-6}$ |
| NTK | Figure 3 (b) | Beta | $10^{-3}$ | $(2, 2)$ | 50 | $10^{-2}$ | Jackson | $(3, 2)$ | $10^{-6}$ |

---

[3] `freealg` is available for installation from PyPI (https://pypi.org/project/freealg), the documentation can be found at https://ameli.github.io/freealg, and the source code is available at https://github.com/ameli/freealg.

Listing 1: A minimal usage example of the `freealg` package.

```python
# Install freealg with "pip install freealg"
import freealg as fa

# Create an object for the Marchenko-Pastur distribution with the parameter λ = 1/50
mp = fa.distributions.MarchenkoPastur(1/50)

# Generate a matrix of size n_s = 1000 corresponding to this distribution
A = mp.matrix(size=1000)

# Create a free-form object for the matrix within the support I = [λ_-, λ_+]
ff = fa.FreeForm(A, support=(mp.lam_m, mp.lam_p))

# Fit the distribution using Jacobi polynomials of degree K = 20, with α = β = 1/2
# Also fit the glue function via Pade of degree [p/q] with p = 1, q = 1.
psi = ff.fit(method='jacobi', K=20, alpha=0.5, beta=0.5, reg=0.0, damp='jackson',
             pade_p=1, pade_q=1, optimizer='ls', plot=True)

# Estimate the empirical spectral density ρ(x), similar to Figure 1(a)
rho = ff.density(plot=True)

# Estimate the Hilbert transform H[ρ](x)
hilb = ff.hilbert(plot=True)

# Estimate the Stieltjes transform m(z) (both branches m^+ and m^-), similar to Figure 2(a,b)
m1, m2 = ff.stieltjes(plot=True)

# Decompress the spectral density corresponding to a larger matrix of size n = 2^5 × n_s,
# similar to Figure 1(c)
rho_large, x = ff.decompress(size=32_000, plot=True)
```

