# OpenReview forum: "Spectral Estimation with Free Decompression"
_NeurIPS.cc/2025/Conference — NeurIPS 2025 spotlight_

### Official Review · Reviewer_GMts · 2025-06-23

**Clarity:** 2
**Significance:** 2
**Originality:** 2
**Rating:** 4
**Confidence:** 2

**Summary:**

This paper studies algorithms that attempt to estimate certain statistics of large matrices that could not even be written down or accessed via matrix-vector product. In this impalpable model, only a small sub-matrix could be accessed by the algorithm, and the main task studied in this paper is to estimate the spectral density. The major idea is a technique called free decomposition, which is essentially the R-transform from random matrix theory. In short, the R-transform of the top-left block could be connected to the R-transform of the whole matrix. Note that this standard result does not restrict the size of the whole matrix, therefore one could view this as an algorithm for the impalpable matrix setting. Of course, computing the R-transform requires Stieltjes transform which the authors shown that could be approximated via a first-order PDE, and they further develop an algorithm to solve this PDE. Finally, experiments are performed to validate the efficiency of the proposed algorithm for large matrix.

**Questions:**

Can you provide an overview of algorithm 1, and subroutines used in it?

There are compilation errors in the pdf, e.g.,

* Line 211, "given by (??)".

**Ethical Concerns:**

["NO or VERY MINOR ethics concerns only"]

**Final Justification:**

After reading authors' detailed responses and other reviewers' comments, I believe authors' contribution, even from an algorithmic RMT perspective is quite significant. Also, authors proposed reasonable clarifications for their algorithm, and I hope they could incorporate them in their revision.

**Limitations:**

Yes.

**Paper Formatting Concerns:**

N/A.

**Quality:**

3

**Strengths And Weaknesses:**

Strengths:

1. Studying algorithm in the impalpable matrix setting is interesting, as modern day machine learning operates on larger and larger matrices. As authors noted, random matrix theory and free probability would be a crucial tool as it studies the asymptotic behavior of random matrices, which could be "inverted" to develop a tool in this setting.

2. The main contribution, in my opinion, is the algorithm developed in section 4 to practically approximate the Stieltjes transform. Several ideas, including finding a different branch of the transform and using Jacobi polynomials to estimate the empirical spectral density, are quite interesting. This could lead to more practical algorithms for free probability and RMT.

Weaknesses:

1. The presentation, especially the first 9 pages, could be significantly improved. Many notations are not defined, e.g., $L^\infty, L^2(\Omega)$ and subroutines in algorithm 1 are mostly left unexplained. This is particularly confusing for algorithm 1, for example, what is EigApprox? What is Glue? How are these two used for computing the Stieltjes transform? Without reading the appendix and digging into the details, it's very hard to grasp. Also, it feels to me that 3.3 is a bit redundant, as it serves as an intuition for the proofs in appendix C, which are not present in the main text anyway. Finally, discussions in section 4 are highly technical, as a reader with some knowledge about RMT, I found it quite hard to understand and follow, in particular section 4.1. In its current form, I feel the paper targets the audience with deep expertise in RMT, not the general audience of NeurIPS.

2. On the other hand, this paper could be treated as an elementary application of RMT and free probability (in particular the R-transform), therefore I feel its theory is not very novel. However, as I've stated above, I'm only a reader with some knowledge about RMT, so reviewers with more expertise in this area should weigh in and judge the particular contribution of the paper. I'm willing to change my ratings based on the opinions from reviewers with more expertise in RMT.

---

> ### Author Rebuttal · Authors · 2025-07-31
>
> We thank the reviewer for their thoughtful review and constructive feedback. We hope the following clarifications fully address the concerns raised and improve the overall clarity and contribution of the manuscript.
>
> ---
>
> ### Clarity and Notation Improvements
>
> We thank the reviewer for their comments regarding notation and presentation. To address these concerns, we have added a new Appendix A that includes:
>
> - A comprehensive nomenclature table defining all commonly used variables and symbols, including previously undefined terms such as $L^2(\Omega)$ (the space of square-integrable functions) and $L^\infty$ (the space of essentially bounded functions).
> - A brief overview of key functional transforms from free probability (Stieltjes transform, Hilbert transform, and R-transform), aimed at readers without prior expertise in random matrix theory.
>
> In the main text, Section 4 now begins with a short paragraph outlining the structure of Algorithm 1. This includes clarifying the purpose of each component and distinguishing which subroutines are essential to understanding the core algorithm, versus those discussed in greater detail in the appendix.
>
> Specifically:
>
> - **EigApprox** refers to the numerical approximation of the smoothed spectral density of the input matrix. Although seemingly routine, this step requires high precision, as the accuracy of the entire pipeline depends on it. We use Chebyshev or Jacobi polynomial expansions to ensure spectral accuracy.
> - **Glue** refers to a procedure inspired by Riemann–Hilbert techniques. It is used to reconstruct the second-sheet branch of the Stieltjes transform from its boundary values, effectively gluing together the analytic continuations across the support interval.
>
> Regarding Section 3.3, we have decided to retain it. Although its role is primarily intuitive, other reviewers found it helpful for navigating the technical content that follows. We believe it supports accessibility rather than introducing redundancy.
>
> We hope these revisions improve the clarity and accessibility of the manuscript for a broader readership, including those less familiar with RMT.
>
> ---
>
> ### Clarifying Theoretical and Practical Contributions
>
> We appreciate the reviewer’s candid reflection and their openness to reconsider their rating. While the foundational properties of the R-transform are indeed classical, our work builds upon these to develop a novel pipeline for estimating the spectral density of large matrices from small submatrices—an inverse problem that, to our knowledge, has not been previously realized in this form.
>
> Our theoretical contributions go beyond a direct application of known results. In particular:
>
> - We develop a nonlinear PDE that characterizes the evolution of the Stieltjes transform under matrix growth. While this PDE can be formally derived from classical properties of the R-transform, we believe its explicit formulation, analysis, and practical use as a reverse compression method—what we term free decompression—are new to the literature.
> - Proposition 4 reveals that every characteristic trajectory necessarily crosses the real axis, which creates a bottleneck in the numerical solution. This insight motivated our use of analytic continuation, an essential step for solving the PDE that we believe is novel in this context.
> - To achieve this continuation, we introduce a numerical procedure inspired by scalar Riemann–Hilbert problems, and leverage gluing two branches and series acceleration techniques (e.g., Wynn’s $\epsilon$-algorithm). These techniques, while classical on their own, are, to our knowledge, applied here for the first time.
> - Appendix E provides a new quantitative analysis (Proposition E.1) showing how errors in the initial spectral approximation propagate along characteristics, complementing the closed-form characteristic solution and polynomial stability bound in Proposition 3.
> - The precision approximation of the Stieltjes transform via Chebyshev or Jacobi polynomial expansions is engineered for accuracy and scalability. The reviewer also acknowledged this as one of the strengths of the paper.
>
> Beyond theory, we also emphasize the practical realization of these ideas. We demonstrate accurate decompression on challenging real-world datasets, including the Facebook graph Laplacian and neural tangent kernels (NTKs), and release an open-source implementation to facilitate further use by the community. To our knowledge, this is the first instance of a free-probability-based decompression method—at any scale—being successfully applied to real-world matrices of this size and complexity.
>
> We hope this clarifies that the paper is not merely an application of known theory, but offers new insights, tools, and practical algorithms that significantly expand the scope of free probability in modern data applications.
>
> ---
>
> We sincerely thank the reviewer for their time and feedback, and would be happy to provide any additional clarification they might find helpful.

---

> > ### Comment · Reviewer_GMts · 2025-08-01
> >
> > I thank authors for the very detailed responses. After reading the rebuttal and other reviewers' reviews, I've decided to raise the score. I hope authors could incorporate their proposed clarifications for the algorithm in the revision.

---

### Official Review · Reviewer_EuBy · 2025-06-24

**Clarity:** 4
**Significance:** 4
**Originality:** 4
**Rating:** 6
**Confidence:** 5

**Summary:**

This paper proposes an algorithm to estimate the spectral density of a very large matrix.

They consider the extreme case of matrices which are too large to be formed in their entirety, either implicitly or explicitly. They call such matrices *impalpable*, and differentiate them from other "less difficult" classes of matrices in Section 2. They work under the assumption that there is an underlying $N \times N$ matrix of which we want to estimate the spectral density, and that one can access an $n \times n$ submatrix of it which is obtained from the intersection of the row and columns corresponding to a randomly sampled an index set. They observe that this assumptions correspond to the assumption that these sub-matrices form a sequence (indexed by $n$, and assuming $N$ is infinite if I have understood correctly) which is asymptotically free.

The key idea of this paper is that this assumption provides access to tools from random matrix theory and free probability theory to extrapolate the spectral density of the $n \times n$ sub-matrix to the full $N \times N$ matrix.

Section 3 introduces these tools. Namely, the Stieltjes transform, its inverse, the R-transfrom, and a theorem of Nica and Speicher which relates the R-transform of a free random matrix to that of a fixed submatrix. This theorem, which tells you how to glean information about the spectral density of a random submatrix of a large full matrix, is known as free compression. The authors propose performing this operation is reverse, to glean information about the spectral density of the full matrix from a smaller random sub-matrix. They call this *free decomposition*.

Directly implementing this operation involves inverting the Stieltjes transform in order to compute the R-transform. This is not generally analytically tractable. The bulk of the technical contribution of this paper is to propose a numerical method to perform free decompression directly using the Stieltjes transform.

Section 3.2 shows that free decomposition can be written as the evolution of a PDE in the Stieltjes transform (Prop. 1), that this can be easily solved (Prop. 2), but that the error may grow "polynomially fast" (Prop. 3), suggesting the need to high precision computation. Section 3.3 provides a very nice informal, intuitive explanation of the previous section.

Section 4 explains some practical challenges of using Propositions 1 and 2 to estimate the Stieltjes transform under free decompression and proposes some practical numerical solutions. Algorithm 1 on page 7 provides psuedo-code for the full proposes algorithm.

Section 5 provides some numerical experiments on synthetic and real data, comparing the time and accuracy of the proposed method.

The authors provide a Python package to implement their algorithm.

**Questions:**

- For the task of computing log-determinants of impalpable matrices, how does your method compare to the FLODANCE algorithm of Ameli et al. (2025)?
- What impact do you imagine this algorithm will have on machine learning? Can you give an example of a machine learning algorithm where this algorithm could replace an existing one to improve its efficiency?

**Ethical Concerns:**

["NO or VERY MINOR ethics concerns only"]

**Final Justification:**

I continue to strongly support this paper.

**Limitations:**

Yes. The limitations are discussed briefly yet comprehensively.

**Paper Formatting Concerns:**

No.

**Quality:**

4

**Strengths And Weaknesses:**

This paper is highly innovative and the authors should be commended for their level of creativity in pulling all these ideas together to provide a solution to a problem which is both highly practically important and which currently has very few tenable solutions. Despite the fact the computing the spectral density of very large "impalpable" matrices is an extremely important problem, before this paper, there were effectively no algorithms to do this.

There are three main innovations which lead to the algorithm: 1) the framing of the matrix as a free random matrix and subsequently framing estimating its spectral density as "free decompression"; 2) describing "free decompression" as a PDE; and 3) coming up with the necessary continuations and smoothings required to numerically solve the PDE. To me, this is an extremely elegant approach and I find it difficult to find fault in it.

The empirical experiments are extensive and compelling.

I believe the algorithm proposed in this paper has the potential to be highly impactful in the field of numerical linear algebra, and may open the possibility of new breakthrough which were previously impossible without such an algorithm.

The paper is extremely well-written. Though technical, the core ideas are presented in an accessible manner and the reader is guided through all of the ideas and innovations in a very clear and helpful way.

---

> ### Author Rebuttal · Authors · 2025-07-31
>
> We thank the reviewer for their enthusiastic and thoughtful feedback. We are delighted that you found the contributions and presentation of our work compelling. Below, we address your comments and provide additional details that we hope you will find helpful.
>
> ---
>
> ### Computing Log-Determinants
>
> To address the reviewer’s question, we conducted a direct numerical comparison between free decompression and the FLODANCE algorithm (Ameli et al., 2025) on the same input matrix used in our experiments (log-NTK from CIFAR-10 with ResNet50). Starting from an initial sub-matrix of size $n_s = 2^{10}$, we estimated the log-determinant $\ell_n = \log\det(\mathbf{A}_n)$ at increasing matrix sizes, up to $n = 50{,}000$, and report the relative errors against ground truth in the table below.
>
>
> | Size $n$           |  $2^{10}$ |  $2^{11}$ |  $2^{12}$ |  $2^{13}$ |  $2^{14}$ |  $2^{15}$ |     $50$K |
> |--------------------|-----------|-----------|-----------|-----------|-----------|-----------|-----------|
> | Free Decompression | $0.001$%  | $0.007$%  | $0.018$%  | $0.016$%  | $0.022$%  | $0.027$%  | $0.193$\% |
> | FLODANCE           | $0.006$%  | $0.283$%  | $0.984$%  | $2.180$%  | $5.401$%  | $5.128$%  | $4.412$\% |
>
> At this range of matrix sizes, free decompression yields consistently more accurate estimates. We attribute this to a key difference in assumptions: FLODANCE relies on the stationarity of a log-normal stochastic process governing the determinants of growing sub-matrices. This assumption typically becomes valid at much larger scales; in the FLODANCE paper, the authors mitigate early-stage deviations by introducing a burn-in period and applying the method only after the stochastic process stabilizes. In contrast, free decompression does not rely on such stationarity and remains accurate even at smaller sizes, provided the spectrum of the initial matrix is large enough to give an accurate approximation of the spectral distribution.
>
> While FLODANCE is well suited to ultra-large matrices where stationarity becomes apparent, free decompression offers a complementary approach that does not rely on such assumptions and remains accurate without requiring burn-in or stochastic modeling. We also note that, using the free decompression PDE, functionals of the distribution can be approximated by ODEs. Consequently, one can derive a continuous-time ODE for the log-determinant $\ell_n$. These types of problems provide rich avenues for future work.
>
> ---
>
> ### Impact and Applications
>
> The ability to reconstruct the full eigenvalue spectrum of a large matrix enables the approximation of a broad class of linear algebraic quantities that appear throughout modern machine learning. We categorize these quantities below according to the *type of spectral functionals* they involve, rather than by application domain, since the same mathematical structures arise in diverse problems.
>
> #### Matrix Functionals
>
> Free decompression (FD) approximates spectral quantities of the form
> $$f(\mathbf{A}_n) = n \int f(\lambda) \rho(\lambda)  \mathrm{d}\lambda,$$
> where $\rho(\lambda)$ is the empirical spectral density of $\mathbf{A}_n$. This framework encompasses a wide class of spectral functionals, including log-determinants $\log \det(\mathbf{A}_n + \sigma^2 \mathbf{I})$, traces of inverse powers such as $\mathrm{tr}(\mathbf{A}_n + \sigma^2 \mathbf{I})^{-1}$ and $\mathrm{tr}(\mathbf{A}_n + \sigma^2 \mathbf{I})^{-2}$, and more generally, any analytic function $f$. These expressions arise in numerous applications, such as:
>
> - Gaussian-process marginal likelihood and predictive variance,
> - Determinantal Point Process (DPP) training and sampling,
> - Laplace-approximated Bayesian inference (e.g., for GLMs and neural networks),
> - PAC-Bayes generalization bounds in deep learning,
> - Kalman filtering and EM-based state-space models.
>
> This category also includes spectral entropy measures such as the Von Neumann entropy,
> $$S_{\mathrm{VN}} = - n \int \lambda \log (\lambda)  \rho(\lambda) \mathrm{d}\lambda,$$
> which appears, for instance, in graph neural network pipelines as a measure of graph complexity or regularization. Extensions to Rényi or Tsallis entropies are similarly supported. Other notable examples include heat kernel traces $\mathrm{tr} e^{-t\mathbf{A}_n}$, log-determinants of Laplacian minors (e.g., spanning-tree counts), and spectral divergences (e.g., KL or Wasserstein) used in domain adaptation. All of these reduce to integrals of the form $\int f(\lambda) \rho(\lambda)  \mathrm{d}\lambda$, and hence fall naturally within the scope of FD. Once $\rho(\lambda)$ is recovered, such functionals can be computed efficiently via one-dimensional quadrature.
>
>
> #### Quadratic Forms
>
> FD can also be used to approximate quadratic forms involving (powers of) inverse matrices:
> $$\boldsymbol{y}^{\intercal} (\mathbf{A}_n + \sigma^2 \mathbf{I})^{-1} \boldsymbol{y}.$$
> While such expressions typically require a full solve, FD can efficiently estimate *expected* values or averages over random probes. This is particularly useful when one requires many such forms (e.g., in kernel ridge regression, influence functions, or dropout variance estimation). For a specific fixed vector $\boldsymbol{y}$, FD is not directly applicable in its standard form, though natural extensions will allow such quantities to be estimated within a modified framework, and are the focus of ongoing work.
>
>
> #### Condition Number Estimation
>
> The condition number $\kappa(\mathbf{A_n}) = \lambda_{\max}/\lambda_{\min}$ can be approximated by extrapolating the spectral edges of $\rho(\lambda)$. FD is effective at recovering the *bulk support* of the spectrum, which can provide reliable bounds in many practical scenarios. However, isolated outlier eigenvalues may not be accurately captured, so combining FD with one or two power iterations is advisable when precise extremal estimates are required. This is particularly relevant in diagnostics of ill-conditioning and edge-of-stability analyses.
>
> ---
>
> We would be happy to include these categories and examples as a short section in the appendix if the reviewer finds this exposition helpful. We welcome any further questions or suggestions the reviewer may have and would be happy to revise the paper accordingly.

---

> > ### Comment · Reviewer_EuBy · 2025-08-04
> >
> > I'd like to thank the authors for their detailed response. I think the content of the response would make a valuable addition to the paper, though my support for the paper is not contingent on this.
> >
> > I'd like to reiterate my strong support for this work which I think will be of great value to the numerical linear algebra and machine learning communities.

---

### Official Review · Reviewer_BF8q · 2025-06-25

**Clarity:** 2
**Significance:** 4
**Originality:** 4
**Rating:** 4
**Confidence:** 4

**Summary:**

This paper introduces "free decompression," a novel method for estimating the spectral density of large impalpable matrices. The method exploits tools from free probability theory, specifically evolving the Stieltjes transform of a small submatrix via a partial differential equation to estimate the spectrum of the full matrix. The paper provides strong theoretical foundations, polynomial error bounds, and justifies the approach on synthetic examples and real-world datasets.

**Questions:**

1) Proposition E.1 in the appendix seems to be one of the critical results and I have several questions here. line 754, what do you mean by first order approximation (is this similar to Taylor/Laurent type expansion and in that case what is the region of validity: radius of convergence, ie. an upper bound on $|\delta m|$)? I can clearly see $m(z)$ is meromorphic, therefore $m$ does have a finite Laurent expansion near the poles (in fact they all seem to be simple poles), however, it is not clear to me what exactly is $\delta m$ and how are you bounding the higher order errors here.
2) line 772: could you prove in detail why $C_y <\infty$ please for all $y$ (in particular why those supremums are finite)? I want to understand how large $C_y$ can get as this is essentially the exponent $\nu.$
3)  How sensitive is the method to the choice of submatrix? Are there better strategies than random sampling?
4) These two questions are not deal-breakers and I understand they could be well out of reach for now: Is there a (non-trivial) lower bound on $\delta m_n$ of polynomial (or poly-log) in $\dfrac{n}{n_s}$? Finally, what about non-hermitian matrices that do occur frequently in practice?

**Ethical Concerns:**

["NO or VERY MINOR ethics concerns only"]

**Final Justification:**

This is a very good paper and I am satisfied with authors response. I am willing to increase my rating if AC deems it necessary.

**Limitations:**

yes

**Quality:**

4

**Strengths And Weaknesses:**

Strengths:
As already mentioned, the paper is thoeretically very strong, including detailed formal propositions and proofs for the evolution equation and error bounds. Unlike purely theoretical work, the authors provide a complete implementation with detailed algorithms and a Python package, making their method more accessible. The experimental validation includes multiple synthetic distributions and challenging real-world applications (for instance Facebook network). Finally, the paper is very well-written with good motivation and helpful visualizations. This paper overall makes a sound contribution to an important problem in numerical linear algebra.

Weaknesses:
The authors emphasizes that despite the limitation of their proposed work to real-world application, there is no competitive alternative. However, such a strong statement probably needs a bit more justification. Could you provide some non-trivial warning signs (i.e. classes of matrices where free decompression is expected to fail except obvious highly-ill conditioned ones) for practitioners and how do the alternative methods compare in such cases? Apart from these, there are a few more things I would like to understand clearly before recommending a full acceptance (please see the questions).

---

> ### Author Rebuttal · Authors · 2025-07-31
>
> Thank you for the careful review of our paper. We are glad to receive your positive appraisal, and we have provided responses to your questions and concerns below.
>
> ---
>
> ### On the Scope and Comparison with Alternatives
>
> Regarding competitive alternatives, the issue is not that we view existing methods as uncompetitive, but rather that we could not find any direct alternatives at all. That is, algorithms that accept a matrix as input and return an approximation of the spectral density of a much larger matrix of the same type, containing the original matrix in its upper-left corner. To our knowledge, the only existing approach is to appeal to limiting cases-e.g., approximating the spectral distribution by a Marchenko-Pastur law. While this can be appealing in some settings, we have found that real datasets are often ill-conditioned and heavy-tailed, requiring more flexible approximations.
>
> Since our method approximates the spectral density, functionals can be estimated directly via quadrature. This makes the FLODANCE algorithm (Ameli et al. 2025) a relevant indirect competitor. A detailed comparison is provided in our response to reviewer *EuBy*, to which we kindly refer the reviewer.
>
> In our internal experiments, we have also identified several challenging matrix classes. The reviewer is correct that the most prominent of these is pathological ill-conditioning, particularly in low-rank matrices. Another difficult case arises when the spectrum has outliers or large spikes caused by structural features of the matrix. Finally, distributions with disjoint support (i.e., nonzero on multiple separate intervals) can also pose difficulties. Each of these cases requires specialized tools that are currently under active development and will be the focus of future work.
>
> ---
>
> ### Clarification on First-Order Approximation
>
> To analyze the stability of the nonlinear PDE, we consider a perturbation of the solution of the form
> $$\tilde{m}(t, z) = m(t, z) + \delta m(t, z),$$
> where $\delta m$ is the first-order variation of $m$. This is not a Taylor or Laurent expansion in $z$, but rather a first-order (in $\delta m$) *functional variation*, corresponding to a linearization of the PDE. Substituting $\tilde{m}$ into the nonlinear PDE and retaining only the terms linear in $\delta m$ yields a linear PDE for $\delta m$—the equation obtained by taking the Gateaux derivative of the original PDE operator, a standard technique in PDE stability analysis. This linearized PDE governs how perturbations to the initial condition $m_0$ propagate over time.
>
> Importantly, this analysis is carried out for $z$ in the upper half-plane $\mathbb{C}^+$, where $m(t, z)$ is a Herglotz function (i.e., holomorphic with positive imaginary part) and has no poles—hence, it is not meromorphic. (Poles arise only on the second Riemann sheet used during the characteristic continuation into the lower half-plane.) As such, notions like Laurent expansion, radius of convergence, and pole structure do not apply, since the Stieltjes transform is analytic on $\mathbb{C}^+$. The perturbation $\delta m$ is likewise defined on $\mathbb{C}^+$ and assumed to remain small in the $\|\cdot\|_\infty$ norm. Proposition E.1 then quantifies how such perturbations evolve, bounding their growth via a Gronwall-type estimate.
>
> ---
>
> ### Clarification on Boundedness of $C_y(t)$ in Proposition E.1
>
> We thank the reviewer for pointing this out, as we agree that further clarification is needed. In Remark E.1, we briefly sketched why $C_y(t)$ in Proposition E.1 remains finite for all $0 < y < \infty$, but we agree that the argument was too coarse and should be made explicit. The key bounds follow from standard properties of Stieltjes transforms of compactly supported densities.
>
> Recall from equation (E.3) that $C_y(t)$ is composed of two terms:
> $$\sup_{z \in \Gamma_y} \left| \Re \frac{\partial_z m}{m^2} \right|, \quad \text{and} \quad \sup_{z \in \Gamma_y} \left| \Im \frac{1}{m} \right|.$$
> We now verify that each of these terms is finite for all $0 < y < \infty$. Let $m(t,z) = \int_{\mathbb{R}} \frac{\rho(t,\xi)}{\xi - z} \mathrm{d}\xi$ with $z = x + i y \in \mathbb{C}^{+}$. Then:
>
> - Since $\rho(t,x)$ is supported in a compact interval $[\lambda_{-}, \lambda_{+}]$ of diameter $D = \lambda_{+} - \lambda_{-}$, we have
> $$|m(t,z)| \geq \Im m(t,z) = \int_{\mathbb{R}} \frac{y \rho(t,\xi)}{(\xi - x)^2 + y^2}  \mathrm{d}\xi \geq \frac{y}{D^2 + y^2} \int \rho(t,\xi) \mathrm{d}\xi = \frac{y}{D^2 + y^2}.$$
> - It follows that $|m(t,z)^{-1}| \leq \frac{D^2 + y^2}{y}$, so $|\Im m(t,z)^{-1}|$ also has this bound.
> - For the derivative, note
> $$\partial_z m(t,z) = \int_{\mathbb{R}} \frac{\rho(t,\xi)}{(\xi - z)^2} \mathrm{d}\xi,$$
> so
> $$|\partial_z m(t,z)| \leq \int_{\mathbb{R}} \frac{\rho(t,\xi)}{(\xi - x)^2 + y^2}  \mathrm{d}\xi \leq \frac{1}{y^2}.$$
> - Combining the above, we obtain
> $$\left| \frac{\partial_z m(t,z)}{m(t,z)^2} \right| \leq \frac{1}{y^2} \left( \frac{D^2 + y^2}{y} \right)^2 = \frac{(D^2 + y^2)^2}{y^4},$$
> which is finite for every $y > 0$.
>
> This confirms that $C_y(t) < \infty$ for all $y > 0$. We will be happy to include this derivation in the revised appendix for completeness.
>
> Note that this bound is not defined at $y = 0$, which is expected: as $y \to 0^+$, the Stieltjes transform approaches the real axis where it develops discontinuities, and quantities like $\partial_z m$ may diverge. However, in both our theory and implementation, we evaluate $m(t, z)$ slightly above the real axis (e.g., at $z = x + i \delta$ for fixed $0 < \delta \ll 1$), where all terms remain finite. In practice, the observed error propagation is significantly milder than this worst-case upper bound suggests.
>
> ---
>
> ### Sampling Considerations
>
> The decompression equation (Theorem 1) holds on the assumption that the submatrix is sampled from a Haar-uniform permutation of the full matrix. That is, the rows and columns are first permuted uniformly at random, and then the top-left $n_s \times n_s$ block is selected. This ensures that the submatrix and its complement are asymptotically free in the large-$n$ limit.
>
> Any form of deterministic, importance-weighted, or leverage-score sampling would break this freeness assumption. Extending the theory to account for such alternatives would require a new analysis to show that the submatrix remains (approximately) free with respect to full matrix, which is an open problem to our knowledge. Therefore, uniform random sampling is not an arbitrary design choice; it *makes the theory valid*.
>
> From a practical standpoint, we find the method is robust to the submatrix sampling. To quantify this, we repeated our experiment on the NTK matrix using *twenty* independent Haar-uniform submatrices, each of size $n_s = 2^{10}$. For each batch, we applied free decompression to reconstruct the spectral density at increasing matrix sizes $n = 2^{11}, 2^{12}, \dots, 2^{15}, 50000$, and compared the result against the exact density at that scale.
>
> We measured the deviation between the estimated and true densities using multiple metrics, including total variation (TV), Jensen–Shannon (JS) divergence, and Kolmogorov–Smirnov (KS) statistic, which take values in the range $[0, 1]$. For each metric, the *mean and standard deviation across batches* are reported below. These results demonstrate that the method is *practically insensitive* to the specific submatrix; even a single random batch provides a sufficiently accurate spectral estimate.
>
> | Size $n$ $\quad$ | TV ($\times 100$) $\quad$| JS ($\times 100$) $\quad$| KS ($\times 100$) $\quad$| $\Delta \mu_1 / \mu_1$ ($\times 100$) | $\Delta \mu_2 / \mu_2$ ($\times 100$) |
> |-----------|-------------------|-------------------|-------------------|-------------------|-------------------|
> | $ 2^{10}$ |   $1.08 \pm 0.08$ |   $3.90 \pm 0.55$ |   $0.42 \pm 0.05$ | $0.33 \pm 0.20$ | $0.31 \pm 0.21$ |
> | $ 2^{11}$ |   $1.07 \pm 0.16$ |   $3.89 \pm 0.76$ |   $0.42 \pm 0.07$ | $0.13 \pm 0.12$ | $0.13 \pm 0.13$ |
> | $ 2^{12}$ |   $2.01 \pm 0.30$ |   $4.69 \pm 0.60$ |   $0.70 \pm 0.12$ | $0.06 \pm 0.04$ | $0.07 \pm 0.04$ |
> | $ 2^{13}$ |   $3.18 \pm 0.51$ |   $6.23 \pm 0.44$ |   $1.15 \pm 0.19$ | $0.02 \pm 0.02$ | $0.02 \pm 0.02$ |
> | $ 2^{14}$ |   $4.26 \pm 0.84$ |   $7.46 \pm 0.80$ |   $1.72 \pm 0.45$ | $0.01 \pm 0.01$ | $0.01 \pm 0.01$ |
> | $ 2^{15}$ |   $5.12 \pm 1.20$ |   $7.88 \pm 1.15$ |   $2.48 \pm 0.82$ | $0.03 \pm 0.01$ | $0.04 \pm 0.02$ |
> | $50$K     |   $5.91 \pm 1.52$ |   $8.26 \pm 1.47$ |   $3.00 \pm 1.13$ |  $0.19 \pm 0.03$ | $0.07 \pm 0.07$ |
>
> ---
>
> ### Regarding a possible lower bound
>
> A universal *lower* bound for $\Vert \delta m(t)\Vert_{L^2 (\Gamma_y)} $ cannot be given: the perturbation $\delta m(0,\cdot)$ can be chosen arbitrarily small (or identically zero), so the sharp bound is simply $\Vert\delta m(t)\Vert_{L^2 (\Gamma_y )} \ge 0$.
>
> Any positive lower bound would depend on additional information about the initial perturbation, not on the PDE itself. For stability and error-propagation purposes the relevant result is therefore the upper (growth) estimate provided in Proposition E.1.
>
> ### Non-Hermitian Matrices
>
> Our current framework is designed specifically for Hermitian matrices. Extending this framework to non-Hermitian matrices (whose spectra lie in the complex plane) is an interesting and important direction. However, doing so would require substantial modifications, including different analytic machinery and stronger structural assumptions. We consider this a valuable but nontrivial extension, and leave it to future work.
>
> ---
>
> We appreciate the reviewer’s constructive feedback and would be happy to provide further clarification or make additional improvements based on any remaining questions or suggestions.

---

> > ### Comment · Reviewer_BF8q · 2025-08-05
> > **Thank you**
> >
> > Thank you for the detailed response/clarification. I maintain my positive outlook on this paper.

---

### Official Review · Reviewer_gtu3 · 2025-07-01

**Clarity:** 3
**Significance:** 2
**Originality:** 3
**Rating:** 4
**Confidence:** 4

**Summary:**

In this paper, the authors consider a problem of spectral estimation, where the goal is to estimate the spectrum of a large matrix from that of a small submatrix. The authors propose an algorithm based on the (time) evolution of the Stieltjes transform. Several mathematical properties of the proposed algorithm are proved, and the performance of the algorithm is tested numerically by synthetic experiments and also several datasets.

**Questions:**

(1) Theoretically, to obtain the spectral density from the Stieltjes transform, one can use the Stieltjes inversion that computes $(1/\pi) \lim_{\epsilon \to 0} \textrm{Im } m(x+i\epsilon)$. I wonder if one can avoid extending the Stieltjes transform to the lower half plane $C^-$ by applying the Stieltjes inversion.

(2) Some random matrices have outlier eigenvalues (for example, due to the presence of underlying low-rank structure), which are crucial in understanding the properties of the random matrices. Are these outlier eigenvalues ignored when approximating the empirical spectral density?

(3) Minor comment: The equation number is missing in line 211.

**Ethical Concerns:**

["NO or VERY MINOR ethics concerns only"]

**Final Justification:**

I thank the authors for the detailed replies. I would keep my scores.

**Limitations:**

The limitations of the work are well addressed. It seems that there would be no negative societal impact of the work.

**Quality:**

2

**Strengths And Weaknesses:**

Strengths:
The problem considered in this paper is of great importance, and the proposed algorithm is novel. The idea is simple, but it has high potential to be developed further since the Stieltjes transform is well-studied object in random matrix theory. The writing is clear in general.

Weaknesses:
Several assumptions, especially one on the freeness, are rather too strong.

---

> ### Author Rebuttal · Authors · 2025-07-31
>
> We thank the reviewer for their thoughtful review and constructive feedback. Below we address the specific questions and concerns raised.
>
> ---
> ### Clarification on Freeness Assumptions
>
> We believe there may be a misunderstanding of the freeness assumption. We do *not* assume that the data matrix is free with a fixed or arbitrary companion matrix. Rather, we rely on a well-established result from free probability: the *asymptotic freeness* of the pair $(\mathbf{E}, \mathbf{P_\sigma A P_\sigma^{-1}})$, where $\mathbf{E}$ is the projection onto the top-left  $n_s \times n_s$ block, $\mathbf{P}_\sigma$ is a Haar-uniform random permutation matrix, and $\mathbf{A}$ is *any* deterministic Hermitian matrix with uniformly bounded operator norm.
>
> Nica (1993, p.5) shows that the random permutation induces asymptotic freeness in the large-$n$ limit. This setting is in fact one of the cleanest regimes in which freeness can be rigorously justified and is widely used in free-probability-based methods, requiring no additional structural assumptions on the data matrix beyond a standard bounded-norm condition.
>
> ---
> ### Analytic continuation and the Stieltjes inversion formula
>
> The reviewer is correct that the Stieltjes inversion formula can be used to obtain the spectral density $\rho(t, x)$ from the Stieltjes transform $m(t, z)$, and this is exactly how we extract the density in the final step of our method (see line 10 of Algorithm 2).
>
> However, the analytic continuation of $m_0(z)$ into the lower half-plane is not related to this inversion step, but is instead required *earlier* in the pipeline: when solving the PDE that evolves $m(t, z)$ from the initial data $m_0(z)$.
>
> In our method, the evolution is performed via the method of characteristics, where the evaluation point $z_0$ flows across the support of the spectral density and enters the lower half-plane. Since $m_0(z)$ is discontinuous across the support, its values in the lower half-plane cannot be obtained from upper half-plane limits alone. For this reason, the analytic continuation plays a crucial role in solving the PDE, rather than in the inversion itself.
>
> More broadly, we spent significant effort exploring several alternative approaches to avoid analytic continuation. In particular, we tried solving the PDE directly with standard finite‐difference and finite‐element schemes: these worked for small values of $t$ but became numerically unstable as $t$ grew. This is because the solutions close to the real line that are used in this inversion formula necessarily rely on information from the lower half-plane. This difficulty is made clear by the form of the solutions in the method of characteristics, and is quantitatively encoded in Proposition 4. Consequently, and despite the need for analytic continuation, the method of characteristics has been the most effective method we have tried in practice. We are also investigating distribution-specific solvers that may bypass this difficulty in certain cases, although such methods are currently limited to special families and remain an area of ongoing work.
>
> We have added these clarifications to the text.
>
> ---
> ### Outlier eigenvalues and low-rank structures
> We also agree with the reviewer that the presence of (almost) low-rank structure and other outlier eigenvalues (such as heavy-tailed behavior) are crucial to understanding the properties of many matrices of interest. Indeed, doing so is a strong motivator for our own study and development of this method.
>
> Our current work focuses on modeling the *bulk of the spectrum*, using the absolutely continuous part of the empirical spectral density. Outliers in the form of isolated eigenvalues without mass (e.g., the top eigenvalue of a finite-rank perturbation) are indeed ignored in our current setting, as they correspond to *measure-zero* features that vanish under the Cauchy integral defining the Stieltjes transform, and therefore do not enter the PDE.
>
> However, *spikes with non-zero mass* (that is, Dirac components in the spectral density) can in principle be handled. The free decompression PDE preserves such components and governs the evolution of both their location and mass. We did not incorporate outliers or spikes into the current work, as their proper treatment involves additional modeling considerations that warrant a dedicated future investigation.
>
> While we have preliminary results in these directions, we believe the present paper already addresses a broad and important class of matrices and is a natural first step in tackling this problem. We will add a short discussion of these extensions in the revised version.
>
> ---
>
> We would again like to thank the reviewer for their constructive feedback. We would be happy to provide any additional clarification the reviewer might find helpful.
>
> ---
>
> #### Reference
>
> [1] Nica, A. (1993). Asymptotically free families of random unitaries in symmetric groups. Pacific Journal of Mathematics, 157(2).

---

### Note · Authors · 2025-08-16

We thank the reviewers and ACs for their thoughtful engagement and constructive discussion. We are pleased that the reviews converged positively, highlighting both the novelty of free decompression and its potential impact. This work introduces a new direction for estimating spectra of very large matrices, with broad applications in numerical linear algebra and machine learning. We hope it can serve as a foundation for future advances at the interface of random matrix theory and large-scale ML.

---

### Decision · Program_Chairs · 2025-09-17

**Decision:**

Accept (spotlight)

**Comment:**

This paper studies the problem of estimating the spectrum of a very large "impalpable" matrix from a small submatrix. They introduce a new method called free decompression to perform this task, based on tools from free probability.

The ideas are novel and interesting, backed up by theoretical foundations, and experiments are performed to validate the efficiency of the proposed algorithm. Many in the NeurIPS audience may not have the background to appreciate the theoretical foundations of the paper, but many would appreciate the problem and woud be interested in learning about these new tools.